# FOXA2 rewires AP-1 for transcriptional reprogramming and lineage plasticity in prostate cancer

Zifeng Wang[1,2,3], Scott L. Townley[4,5], Songqi Zhang [1,2], Mingyu Liu[1,2], Muqing Li[1,2], Maryam Labaf[1,6], Susan Patalano[1,2], Kavita Venkataramani[2], Kellee R. Siegfried [2], Jill A. Macoska [1,2], Dong Han [1,2], Shuai Gao[7,8], Gail P. Risbridger [9,10,11,12,13], Renea A. Taylor [9,11,12,13,14], Mitchell G. Lawrence [9,10,11,12,13], Housheng Hansen He [15,16], Luke A. Selth [4,5,17] & Changmeng Cai [1,2] ✉

FOXA family proteins act as pioneer factors by remodeling compact chromatin structures. FOXA1 is crucial for the chromatin binding of the androgen receptor (AR) in both normal prostate epithelial cells and the luminal subtype of prostate cancer (PCa). Recent studies have highlighted the emergence of FOXA2 as an adaptive response to AR signaling inhibition treatments. However, the role of the FOXA1 to FOXA2 transition in regulating cancer lineage plasticity remains unclear. Our study demonstrates that FOXA2 binds to distinct classes of developmental enhancers in multiple AR-independent PCa subtypes, with its binding depending on LSD1. Moreover, we reveal that FOXA2 collaborates with JUN at chromatin and promotes transcriptional reprogramming of AP-1 in lineage-plastic cancer cells, thereby facilitating cell state transitions to multiple lineages. Overall, our findings underscore the pivotal role of FOXA2 as a pan-plasticity driver that rewires AP-1 to induce the differential transcriptional reprogramming necessary for cancer cell lineage plasticity.

Prostate cancer (PCa) has the ability to evade standard androgen deprivation therapies (ADTs) and progress to castration-resistant PCa (CRPC)[1]. Although CRPC can be further treated with second-generation androgen receptor (AR) signaling inhibition (ARSi) agents, such as abiraterone, enzalutamide, apalutamide, and daralutamide[2–5], tumors eventually become resistant, with a subset transitioning to an AR-independent subtype[6]—such as neuroendocrine PCa (NEPC)—via lineage plasticity[7–9]. Treatment-induced NEPC is

[1]Center for Personalized Cancer Therapy, University of Massachusetts Boston, Boston, MA 02125, USA. [2]Department of Biology, University of Massachusetts Boston, Boston, MA 02125, USA. [3]Yale Stem Cell Center, Department of Cell Biology and Department of Genetics, Yale University School of Medicine, New Haven, CT 06510, USA. [4]Flinders University, College of Medicine and Public Health, Flinders Health and Medical Research Institute, Bedford Park, SA 5042, Australia. [5]Freemasons Centre for Male Health and Wellbeing, Flinders University, Bedford Park, SA 5042, Australia. [6]Department of Mathematics, University of Massachusetts Boston, Boston, MA 02125, USA. [7]Department of Cell Biology and Anatomy, New York Medical College, Valhalla, New York 10595, USA. [8]Department of Biochemistry and Molecular Biology, New York Medical College, Valhalla, New York 10595, USA. [9]Melbourne Urological Research Alliance, Biomedicine Discovery Institute, Monash University, Melbourne, VIC 3800, Australia. [10]Department of Anatomy and Developmental Biology, Biomedicine Discovery Institute, Cancer Program, Monash University, Melbourne, VIC 3800, Australia. [11]Cancer Research Division, Peter MacCallum Cancer Centre, Melbourne, Victoria 3000, Australia. [12]Sir Peter MacCallum Department of Oncology, University of Melbourne, Victoria 3010, Australia. [13]Cabrini Institute, Cabrini Health, Malvern, VIC 3144, Australia. [14]Department of Physiology, Biomedicine Discovery Institute, Cancer Program, Monash University, Melbourne, VIC 3800, Australia. [15]Department of Medical Biophysics, University of Toronto, Toronto, ON M5G1L7, Canada. [16]Princess Margaret Cancer Center, University Health Network, Toronto, ON M5G1L7, Canada. [17]Adelaide Medical School, Faculty of Health and Medical Sciences, The University of Adelaide, Adelaide, SA 5000, Australia. ✉e-mail: changmeng.cai@umb.edu

characterized by genomic and epigenetic features including *MYCN* amplification, *RB1* loss, *TP53* deletions/mutations, and the expression of neuroendocrine (NE) markers[10–12]. In addition to NEPC, CRPC can also progress to other molecular subtypes such as double-negative PCa (DNPC), characterized by low/negative AR and NE marker expression[13]. However, the mechanisms driving PCa lineage plasticity remain poorly understood, and treatment options for these tumors are limited[6,7]. Therefore, there is an urgent need for a better understanding of the mechanism underlying lineage plasticity and the identification of novel therapeutic targets.

FOXA (Forkhead Box A) protein family members, including FOXA1, 2, and 3, are master pioneer transcription factors that play a crucial role in loosening the compact chromatin structure and facilitating the binding of other transcription factors, thus regulating tissue-specific gene expression[14,15]. In prostate adenocarcinoma, FOXA1 acts as a critical pioneer factor for AR, and the FOXA1/AR signaling pathway is essential for the development of this tumor subtype[16–18]. Moreover, FOXA1 is frequently mutated and altered in CRPC and these gain-of-function alterations can promote resistance to ARSi[19–22]. Interestingly, while FOXA1 expression is significantly reduced in NEPC, FOXA2 expression is markedly increased[23], and recent studies have suggested that FOXA2 promotes the progression of NEPC[24,25]. However, the precise molecular function and regulation of FOXA2 in PCa remain unclear, particularly in terms of its pioneer factor activity and its interaction with cooperative transcription factors.

In our recent reports[26], we discovered that LSD1 (Lysine-Specific Demethylase 1 or KDM1A), a well-known lysine demethylase for mono- or di-methylated histone 3 lysine 4 (H3K4me1,2)[27,28], acts as a binding partner of FOXA1 at enhancer regions[29]. LSD1 stabilizes FOXA1 chromatin binding globally in PCa cells by demethylation of lysine 270 (K270) of FOXA1, a critical amino acid adjacent to the wing2 loop (aa 247-269) of the Forkhead DNA binding domain (FKHD)[26,29]. This FOXA1-LSD1 complex facilitates AR chromatin binding and promotes its transcriptional program in luminal subtype of PCa. Interestingly, FOXA2 shares almost identical amino acid sequences at the wing2 loop (amino acids 242-264) and has a lysine (K265) adjacent to this region. Therefore, FOXA2 chromatin binding may be similarly regulated by LSD1-mediated demethylation.

In this work, through examining a series of AR-independent CRPC models, we reveal that FOXA2 exhibits different chromatin binding landscapes, controlling the chromatin accessibility of distinct classes of developmental enhancers in CRPC tumor subtypes, with its chromatin binding being dependent on LSD1-mediated demethylation. Importantly, we identify JUN as a major cooperative transcription factor of FOXA2: chromatin binding of the JUN-containing AP-1 complex depends on the pioneer factor activity of FOXA2, enabling AP-1 to activate lineage-specific enhancers and reprogram the transcriptional networks in AR-independent CRPC. Moreover, we demonstrate that FOXA2 overexpression in AR-dependent luminal PCa cells may initiate tumor progression to a multilineage transition state. Overall, our study indicates that FOXA2 drives PCa lineage plasticity through transcriptional reprogramming of AP-1.

## Results

### FOXA2 binds to distinct classes of enhancers in various molecular subtypes of AR-null CRPC

Previous studies using metastatic tumors from same patients have identified lineage reprogramming in CRPC[30]. In collaboration with MURAL (Melbourne Urological Research Alliance), we interrogated a pair of PDX models, PDX-201.1A-Cx and PDX-201.2A-Cx (referred to as 201.1 and 201.2 herein), which were generated from dura and lung metastases from the same patient who progressed on androgen deprivation therapy (ADT), AR signaling inhibitors (abiraterone, enzalutamide), and taxanes (docetaxel, cabazitaxel) (Fig. 1a)[30,31]. The 201.1 model is typical AR-driven PCa (ARPC) with high expression of

FOXA1 and gain-of-function AR mutations, whereas 201.2 is a DNPC model that has high FOXA2 and low FOXA1 expression[30] (Fig. 1b). By comparing the 201.1 and 201.2 transcriptomes, we found that genes overexpressed in the 201.1 model are enriched for AR signaling while 201.2 overexpressed genes are enriched for mesenchymal/basal cell maintenance, FGF pathway, and WNT signaling (Fig. 1c). ChIP-seq analyses revealed that the 201.2 FOXA2 cistrome is highly distinct from the 201.1 FOXA1 cistrome (Fig. 1d). Importantly, using Binding and Expression Target Analysis (BETA)[32], we demonstrated that the differentially overexpressed genes in 201.1 and 201.2 models were strongly associated with FOXA1 or FOXA2 binding, respectively (Fig. 1e, f), suggesting that FOXA1/2 are key drivers of their lineage-dependent transcription programs. These analyses also predicted the potential direct targets of FOXA1 and FOXA2: FOXA1 targets were enriched for epithelial cell and prostate development as well as AR signaling while FOXA2 targets were enriched for WNT pathway, cell morphogenesis, and neurogenesis (Fig. 1g).

Examining common PCa cell line models, we found that FOXA2 protein was expressed in a DNPC model, PC-3, and an NEPC model, NCI-H660, but not expressed in adenocarcinoma PCa cell lines, LNCaP or CWR-22Rv1 (Fig. 2a). To further investigate the chromatin activities of FOXA2 in AR-null prostate cancer, we performed ChIP-seq analyses of FOXA2 in PC-3 and NCI-H660 models and compared these data with our previously published FOXA1 ChIP-seq results in LNCaP cells[26]. Similar to the PDX models, FOXA2 binding sites in these cells were also distinct to the FOXA1 binding (Fig. 2b). Significantly, we also identified three unique classes of FOXA2 binding sites: class 1 specific for PC-3, class 2 specific for NCI-H660, and class 3 specific for 201.2 (Fig. 2c). These classes of sites were highly correlated with open/transcriptionally active chromatin in each model, as determined by ATAC-seq or H3K27ac ChIP-seq, indicating they are active enhancers. Examining functional enrichment at genes related to these clustered sites, we found that genes in class 1 are enriched for mesenchymal cell differentiation and branching morphogenesis, genes in class 2 are enriched for neuronal development, and genes in cluster 3 are enriched for WNT pathway (Fig. 2d). These data suggest that FOXA2 binds to distinct classes of developmental enhancers in AR-independent CRPC and that this binding is associated with increased chromatin accessibility at these loci.

A recent study reported a molecular subtyping strategy of CRPC using chromatin accessibility signatures determined by ATAC-seq as indicators of chromatin openness and active enhancers[33]. This study defines CRPC into AR, SCL (stem cell-like), NE (neuroendocrine), and WNT subtypes with specific signatures of ATAC signal in each subtype. Using this method, SU2C mCRPC patient samples[34] could be stratified into the four subtypes. As shown in Fig. 2e, *AR* and *FOXA1* expression levels were significantly decreased while *FOXA2* expression was increased in all three AR-independent CRPC subtypes. We next applied the ATAC signatures in our five PCa models (using H3K27ac as an alternative to ATAC for 201.1 and 201.2). As expected, LNCaP cells have enrichment of ATAC signal and enhancer markers at the CRPC-AR class (Fig. 2f). AR and FOXA1 were co-occupied at these sites, consistent with the role of FOXA1 as a pioneer factor in regulating AR signaling and the adenocarcinoma phenotype. In PC-3 cells, ATAC signal and active histone markers were enriched in CRPC-SCL sites and FOXA2 was predominantly enriched in this class of binding sites. For the NCI-H660 model, as expected, the enrichment of active chromatin markers and FOXA2 were found in the CRPC-NE class. For the PDX models, while 201.1 is strongly associated with the CRPC-AR subtype, which was expected given it is dependent on AR and exhibits an adenocarcinoma phenotype, the DNPC model 201.2 exhibited enrichment of H3K27ac and FOXA2 in the WNT class, suggesting that it belongs to the CRPC-WNT subtype. Multiple FOXA2 antibodies were tested for these analyses, and similar results were obtained (Supplementary Fig. 1a–e). Overall, these data demonstrate that FOXA2 binds to key lineage-specific enhancers associated with major AR-independent CRPC

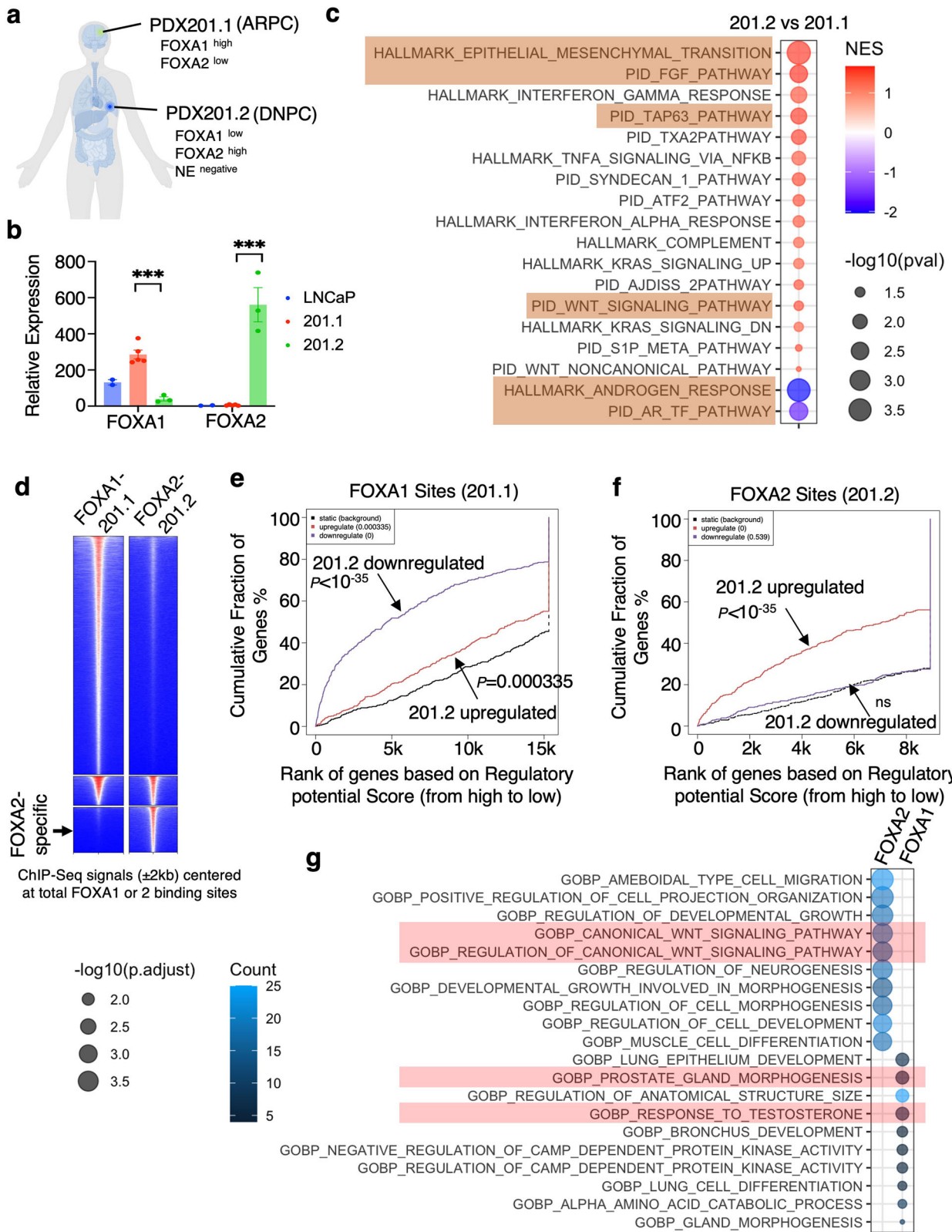

**Fig. 1 | The switch from FOXA1 to FOXA2 promotes lineage reprogramming of CRPC. a** PDX-201.1A-Cx (201.1) and PDX-201.2A-Cx (201.2) were derived from distinct metastases from a single patient (201.1—dura; 201.2—lung). **b** Relative expression of FOXA1 and FOXA2 in these PDXs and LNCaP cells based on bulk RNA-seq data (201.1 $n = 4$ independent tumor samples; 201.2 $n = 3$ independent tumors; data represented as mean ± SEM; statistical significance determined by unpaired two-sided $t$-test). **c** GSEA for transcriptomes (RNA-seq) of PDX201.2 versus PDX201.1. **d** Heatmap view for the ChIP-FOXA1 or FOXA2 centered at the FOXA1 and FOXA2 binding sites in these two PDX models. The intensity of the colors represents the signal strength, with red indicating a higher signal and blue indicating a lower signal. **e**, **f** BETA integrating ChIP-FOXA1 peaks in 201.1 (**e**) or ChIP-FOXA2 peaks in 201.2 (**f**) with RNA-seq data of PDX201.2 versus PDX201.1. **g** Gene ontology (GO) annotation for potential direct targets of FOXA1 in PDX201.1 and FOXA2 in PDX201.2 (identified from BETA). ns ($P > 0.05$), *($0.01 < P < 0.05$), **($0.001 < P < 0.01$), ***($P < 0.001$), and ****($P < 0.0001$) were used to indicate the levels of $P$-value. Source data are provided as a Source Data file. **a** Created with BioRender.com released under a Creative Commons Attribution-NonCommercial-NoDerivs 4.0 International license https://creativecommons.org/licenses/by-nc-nd/4.0/deed.en.

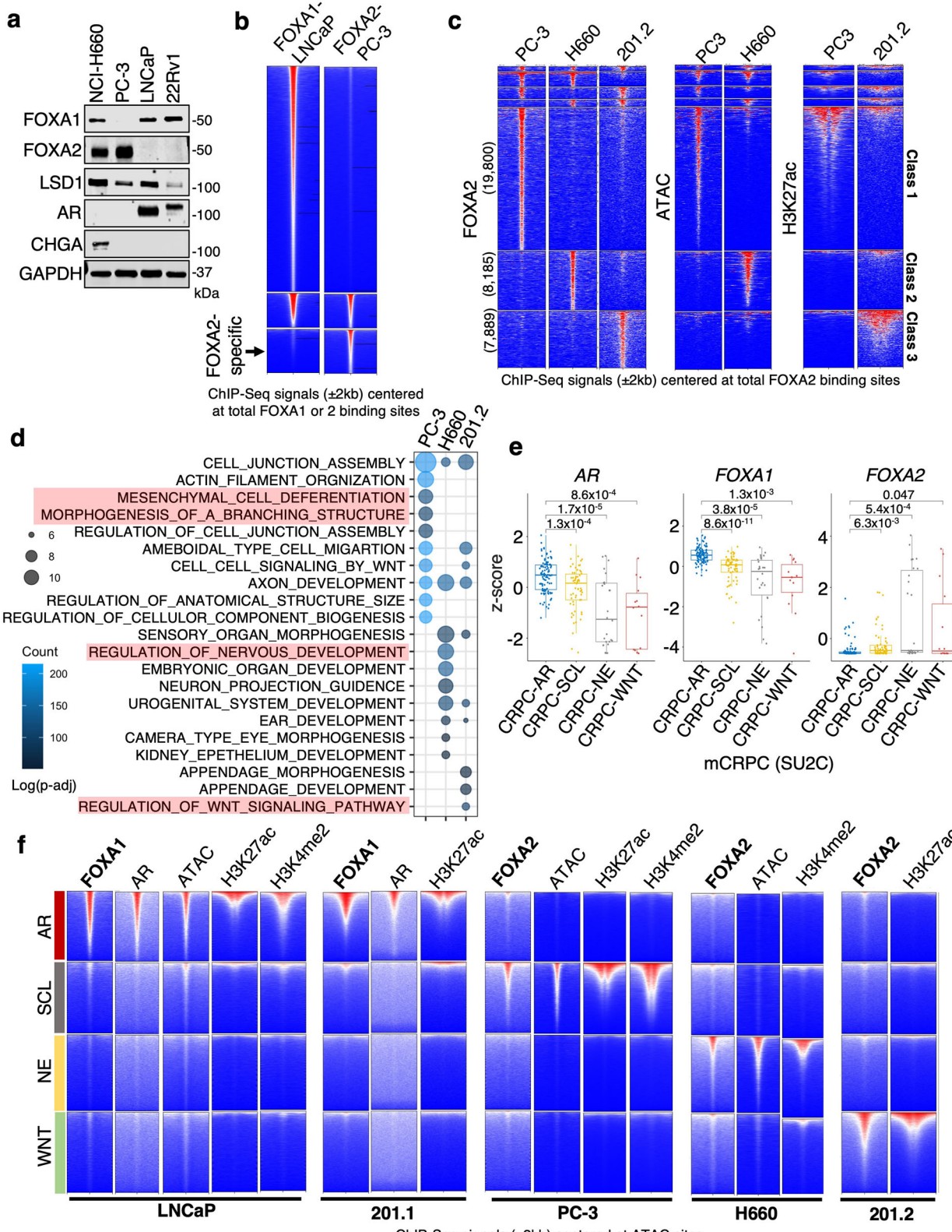

**Fig. 2 | FOXA2 binds to distinct classes of enhancers in three molecular subtypes of AR-independent CRPC. a** Immunoblotting for indicated proteins in PCa cell lines (*n* = 3 independent experiments). **b** Heatmap view for the ChIP-FOXA1 or FOXA2 centered at the FOXA1 and FOXA2 sites in LNCaP or PC-3 cells. **c** Heatmap view for the peaks of FOXA2, ATAC, or H3K27ac in PC-3, NCI-H660, or 201.2 models centered at the FOXA2 sites. **d** GO annotation was performed on genes associated with FOXA2 peaks that are unique to PC-3 (class 1), NCI-H660 (class 2), and 201.2 (class 3) models. **e** Boxplot of mRNA expression (z-score) of *AR*, *FOXA1*, and *FOXA2*

in previously defined CRPC subtypes—AR, SCL, NE, and WNT, using SU2C mCRPC cohort (CRPC-AR, *n* = 104; CRPC-SCL, *n* = 62; CRPC-NE, *n* = 26; CRPC-WNT, *n* = 14; center: median; box: 25th–75th IQR; whiskers: 1.5x IQR; outliers: individual data points; statistical significance determined by unpaired two-sided *t*-test). **f** Heatmap view for the ChIP-seq signal of indicated proteins centered at specific chromatin sites exhibiting different ATAC signatures for CRPC subtypes. ns (*P* > 0.05), *(0.01 < *P* < 0.05), **(0.001 < *P* < 0.01), ***(*P* < 0.001), and ****(*P* < 0.0001) were used to indicate the levels of *P*-value.

subtypes, suggesting that it may have a role in regulating distinct CRPC transcriptional programs.

## FOXA2 chromatin binding and transcription program are dependent on the demethylase activity of LSD1

The chromatin binding of FOXA2 was highly associated with chromatin accessibility, enhancer activation, and enhancer-associated histone modifications (Fig. 3a and Supplementary Fig. 2a). More importantly, it was strongly associated with LSD1 chromatin binding and co-enriched at the subtype-specific ATAC sites of AR-null cells (Fig. 3b). Since FOXA1 protein is an LSD1 substrate and given the similarity between the wing2 loops in FOXA1 and FOXA2[26], we hypothesized that FOXA2 may also be regulated by LSD1. To address this hypothesis, we first evaluated FOXA2 chromatin binding by ChIP-seq in response to two

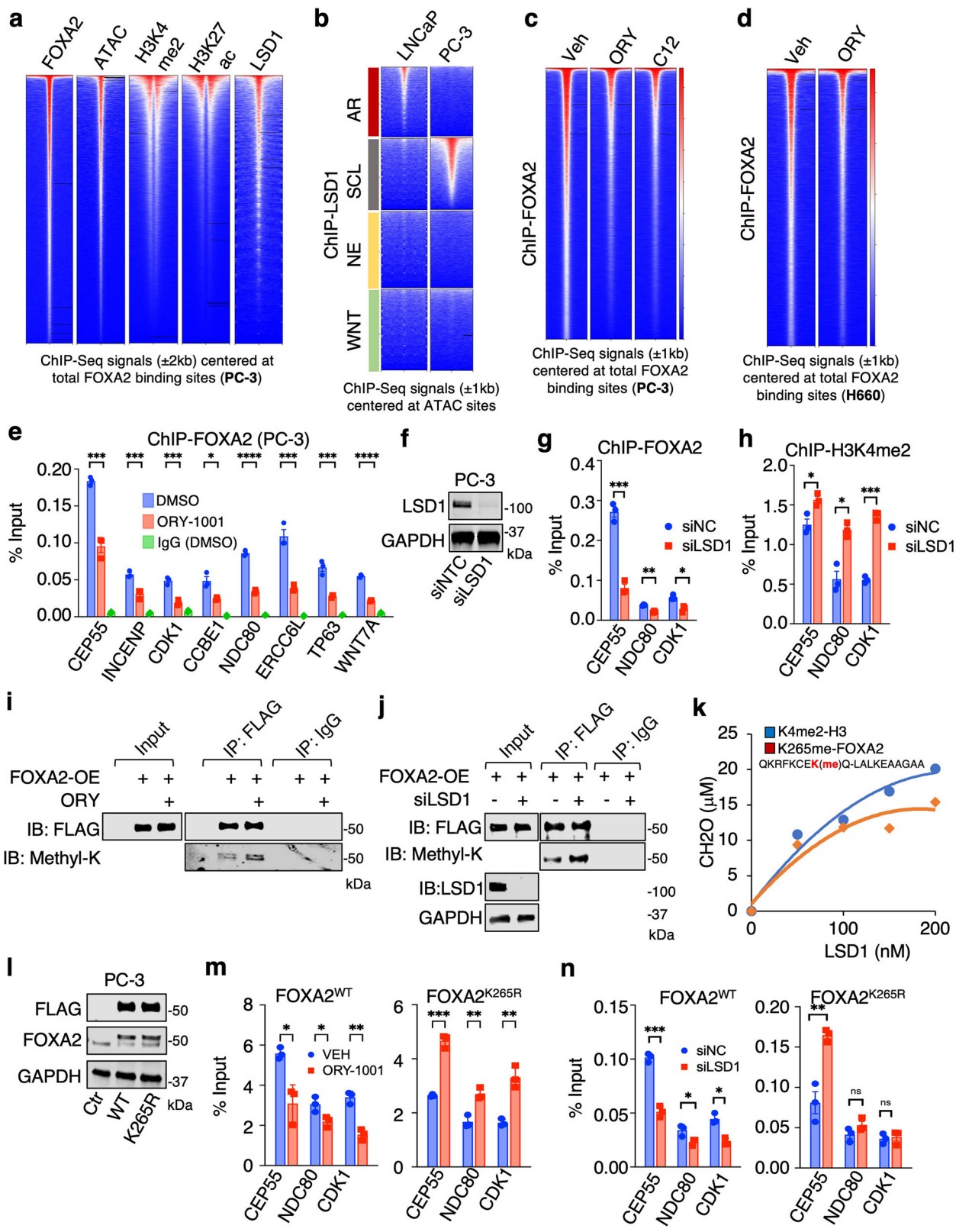

**Fig. 3 | FOXA2 chromatin binding is promoted by LSD1. a** Heatmap view for FOXA2, ATAC (Assay for Transposase-Accessible Chromatin using sequencing), H3K4me2, H3K27ac, and LSD1 ChIP-seq signal intensity at FOXA2 binding sites in PC-3 cells. **b** Heatmap view for the ChIP-seq signal of LSD1[29,61] centered at specific chromatin sites exhibiting different ATAC signatures. **c** Heatmap view of FOXA2 ChIP-seq signal in PC-3 cells treated with vehicle or LSD1 inhibitors, ORY-1001 (10 μM) or C12 (0.5 μM), for 4 h. **d** Heatmap view for FOXA2 signal in NCI-H660 cells treated with vehicle or ORY-1001(10 μM for 4 h). **e** ChIP-qPCR for FOXA2 binding at indicated FOXA2 target sites ($n = 3$ independent samples; data represented as mean ± SEM; statistical significance determined by unpaired two-sided $t$-test). **f** Immunoblotting for LSD1 in PC-3 cells transfected with siRNAs against non-target control (NTC) or LSD1 ($n = 3$ independent experiments). **g, h** ChIP-qPCR for FOXA2 binding (**g**) and H3K4me2 levels (**h**) at indicated FOXA2 target sites ($n = 3$ independent samples; data represented as mean ± SEM; statistical significance determined by unpaired two-sided $t$-test). **i, j** Immunoblotting for methyl-lysine on immunopurified proteins from FLAG-tagged FOXA2 expressing PC-3 cells treated with vehicle or ORY-1001(10 μM) for 24 h (**i**) or transfected with siNC or siLSD1 (**j**) ($n = 3$ independent experiments). **k** In vitro demethylation assay using synthetic H3K4me2 peptide (1–21 aa) or K265-methylated FOXA2 peptide (258-276aa) as substrates incubated with recombinant LSD1 proteins. **l** PC-3 cell lines stably expressing control vector, 3xFLAG-tagged FOXA2-WT, or 3xFLAG-tagged K265R mutant (FOXA2$^{WT}$ or FOXA2$^{K265R}$ cells) were established. Immunoblotting for indicated proteins in these stable lines ($n = 3$ independent experiments). **m, n** ChIP-qPCR for FLAG or FOXA2 binding at the target sites was performed in these stable cell lines treated with vehicle or ORY-1001(10 μM) for 4 h (**m**) or transfected with/out siLSD1 (**n**), respectively ($n = 3$ independent samples; data represented as mean ± SEM; statistical significance determined by unpaired two-sided $t$-test). ns ($P > 0.05$), *($0.01 < P < 0.05$), **($0.001 < P < 0.01$), ***($P < 0.001$), and ****($P < 0.0001$) were used to indicate the levels of $P$-value. Source data are provided as a Source Data file.

potent LSD1 inhibitors, ORY1001[35] or C12[36]. As shown in Fig. 3c, d, LSD1 inhibition globally disrupted the association of FOXA2 with chromatin in PC-3 and NCI-H660 cells, an effect that was further confirmed by examining FOXA2 DNA binding at a panel of target sites by inhibiting or silencing LSD1 (Fig. 3e–h). Having confirmed that LSD1 is required for FOXA2 chromatin binding, we next examined methylation of FOXA2 by LSD1. Indeed, we observed a substantial increase in the levels of methylated FOXA2 without affecting FOXA2 protein levels when LSD1 was inhibited or silenced in PC-3 cells overexpressing FLAG-tagged FOXA2 (Fig. 3i, j). The K265 amino acid residue exists in a region of FOXA2 that is highly conserved with a domain in FOXA1 that contains K270, a lysine that is demethylated by LSD1. Using mass spectrometry, we confirmed the presence of FOXA2 K265 methylation (Supplementary Fig. 2b) and demonstrated that the peptide containing monomethylated K265 can be directly demethylated by the recombinant LSD1 protein (Fig. 3k). To further determine if LSD1 regulates FOXA2 binding through K265 demethylation, we generated a methylation-deficient mutant, FOXA2-K265R (Fig. 3l). As shown in Fig. 3m, n, chromatin binding of the mutant FOXA2 was no longer impaired by LSD1 inhibition, suggesting that K265 methylation is required for the enhancing effect of LSD1 on FOXA2 binding.

We further performed RNA-seq analyses in PC-3 and NCI-H660 cell lines following the silencing of FOXA2 and the inhibition or silencing of LSD1. We observed that silencing FOXA2 did not restore AR signaling in these AR-independent cell lines (Supplementary Fig. 3a–c). Our results, demonstrated in Fig. 4a, reveal that genes activated by LSD1 (downregulated by LSD1-i or LSD1-siRNA) are enriched in oncogenic programs that are similar to those activated by FOXA2. Notably, the transcription activation program driven by FOXA2 was significantly impaired when LSD1 was inhibited or silenced (Fig. 4b–e). Furthermore, the inhibition or silencing of LSD1 led to a reduction in the transcription signatures of the SCL lineage in PC-3 cells and the NE lineage in NCI-H660 cells (Supplementary Fig. 3d–g).

To further investigate whether LSD1 inhibition can block the tumor-promoting activities of FOXA2, we first examined the function of FOXA2 in enabling growth and other oncogenic features of the PC-3 model. As shown in Fig. 4f–i and Supplementary Fig. 4a, b, silencing FOXA2 expression decreased PC-3 cell growth, migration, and invasion in vitro. Consistent with these phenotypes, LSD1 inhibition can block the activation of Aurora kinase pathway (Fig. 4j), which is maintained by FOXA2 in PC-3 cells (Fig. 4f). Significantly, elevated proliferation and migration of PC-3 cells induced by overexpression of FOXA2 were repressed by LSD1 inhibition (Fig. 4k, l and Supplementary Fig. 4c–d), demonstrating the oncogenic interplay between these two factors. Providing further evidence for such interplay, both FOXA2 knockdown (Supplementary Fig. 4e) and treatment with the phase 2 LSD1 inhibitor, ORY1001, in the PC-3 model, markedly repressed in vivo xenograft tumor growth in mice and metastasis in zebrafish embryos (Fig. 4m–p).

Similarly, FOXA2 and LSD1 also regulated cell proliferation in the NCI-H660 model (Supplementary Fig. 4f, g). Altogether, these results demonstrate that the chromatin binding, transcription program, and tumor-promoting function of FOXA2 are strongly regulated by LSD1.

## FOXA2 functions as a major pioneer factor of JUN

Motif enrichment analysis of FOXA2 binding sites revealed a significant enrichment of JUN (also called C-Jun) and other AP-1 factors as well as MAF in PC-3 cells, whereas motifs recognized by these factors were not enriched in the FOXA1 cistrome from LNCaP cells (Supplementary Fig. 5a). Similarly, the JUN motif was enriched in the 201.2 FOXA2 cistrome in addition to motifs bound by SOX, ONECUT, NEUROD1 and LEF1 proteins, whereas the 201.1 FOXA1 cistrome exhibited selective enrichment of XFD-1, HCM1 and HOX motifs (Supplementary Fig. 5b). Many of these factors enriched in FOXA2 sites have been implicated in CRPC lineage plasticity[37]. Consistently, the JUN motif was enriched at FOXA2 sites in all three AR-independent PCa models (Fig. 5a).

Interestingly, while JUN and JUND were both overexpressed in DNPC and NEPC models, the expression of FOS proteins, which are dimeric partners of JUN proteins, appears to be tumor-type dependent (Fig. 5b). To further investigate the potential interplay between FOXA2 and AP-1 factors, we performed ChIP-seq of JUN in the PC-3 and H660 models and FOSL1 in the PC-3 model. As shown in Fig. 5c and Supplementary Fig. 5c-d, FOXA2 chromatin binding was significantly associated with JUN and FOSL1 binding in these models. Using co-immunoprecipitation assays, we also found that AP-1 factors can interact with FOXA2 and LSD1 (Fig. 5d, e and Supplementary Fig. 6a). Given the strong association between FOXA2 and JUN binding, we hypothesized that FOXA2 may function as a pioneer factor of JUN. Indeed, silencing FOXA2 led to a dramatic decrease in JUN binding in both PC-3 and NCI-H660 models and a decrease in FOSL1 binding in PC-3 cells (Fig. 5f, g). This enhancing effect on JUN binding was validated at several FOXA2-bound chromatin sites by FOXA2 silencing or overexpression (Fig. 5h and Supplementary Fig. 6b). In contrast to these AR-independent models, JUN binding in the AR-dependent LNCaP model was not decreased by FOXA1 silencing (Fig. 5i) and no strong interaction between FOXA1 and JUN was detected by co-immunoprecipitation (Supplementary Fig. 6c).

We next sought to determine how FOXA2 interacts with JUN. By overexpressing different domains of FOXA2 in cells, we found that the region containing the Forkhead DNA binding domain (166-324aa) may be responsible for its interaction with JUN (Supplementary Fig. 6d). Moreover, re-ChIP assays demonstrated that JUN can form a complex with FOXA2 at FOXA2 target sites (Supplementary Fig. 6e). Within the co-binding sites, FOXA2 and JUN motifs are often adjacent to each other, and the most common composite motif is a 5′ FOXA2 motif (11nt) immediately followed by a JUN motif (8nt) (Fig. 5j and Supplementary Fig. 6f-h).

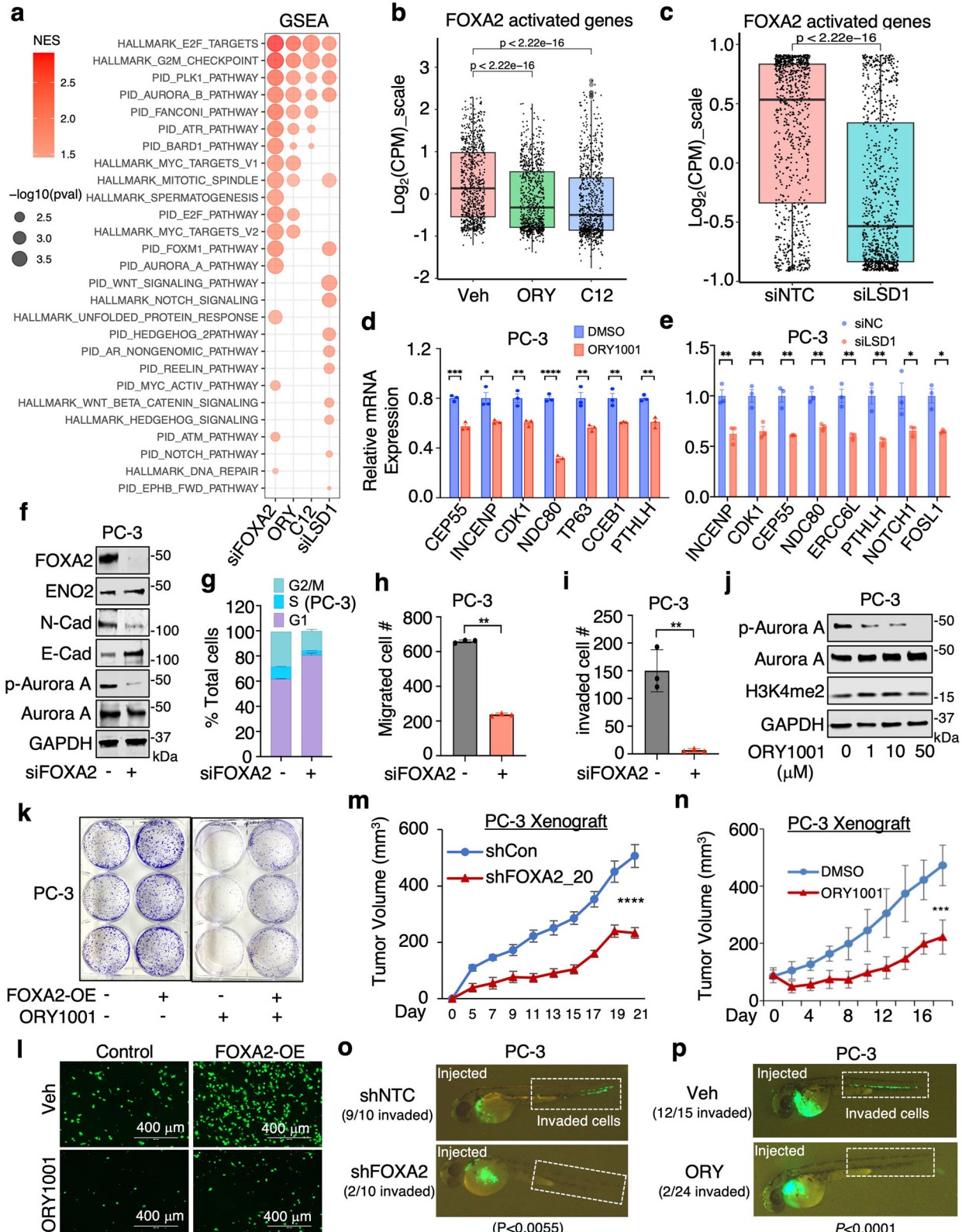

We next determined whether the chromatin binding of AP-1 is associated with lineage-specific sites of FOXA2. As shown in Fig. 5k, the binding sites of JUN/FOSL1 in PC-3 cells were tightly associated with FOXA2 at previously identified class 1 (PC-3 specific) sites while JUN binding sites in NCI-H660 cells were associated with FOXA2 at class 2 (H660-specific) sites (see Fig. 2c). We then examined whether JUN binding is also lineage specific in CRPC subtypes. In the LNCaP model,

JUN was not enriched in any of these classes, suggesting that JUN does not have a role in lineage maintenance of CRPC-AR tumors (Fig. 5l). In the PC-3 model, JUN/FOSL1 were predominantly enriched in the SCL class, and in the NCI-H660 model, the enrichment of JUN was found in the NE class, indicating the possible cooperative binding of AP-1 with FOXA2 in CRPC cells. Overall, these data indicate that FOXA2 and JUN cooperatively bind to distinct classes of enhancers in AR-null CRPC subtypes.

**Fig. 4 | FOXA2 transcription activity and tumor-promoting function are regulated by LSD1.** **a** Gene Set Enrichment Analysis (GSEA) was performed on differential expression results (obtained from RNA-seq) for siNTC versus siFOXA2, vehicle versus ORY-1001 (10 μM, 24 h), vehicle versus C12 (0.5 μM, 24 h), and siNTC versus siLSD1 in PC-3 cells using HALLMARK and PID datasets. Red dots indicate enriched pathways for downregulated genes by FOXA2 silencing or LSD1 inhibition/silencing. **b**, **c** Boxplot view for the expression of FOXA2 activated genes ($n = 834$ genes, $\log_2$(fold-change)>1 & $P < 0.05$, $\log_2$(CPM_zscore)) in PC3 cells treated with/out LSD1 inhibitors (**b**) or transfected with/out siLSD1 (**c**) (center: median; box: 25th to 75th IQR; whiskers: 1.5x IQR; outliers: individual data points; statistical significance determined by unpaired two-sided $t$-test). **d**, **e** qRT-PCR analyses for indicated direct FOXA2 targets in PC-3 cells treated with either vehicle or ORY-1001 (10 μM, 24 h) (**d**) or transfected with/out siLSD1 (**e**) ($n = 3$ independent samples; data represented as mean ± SEM; statistical significance determined by unpaired two-sided $t$-test). **f** Immunoblotting for indicated proteins in PC-3 cells transfected with siFOXA2 versus siNTC ($n = 3$ independent experiments). **g**–**i** Cell cycle analysis (**g**), transwell migration assay (**h**), or Boyden chamber invasion assay (**i**) in PC-3 cells transfected with siFOXA2 versus siNTC ($n = 3$ independent samples; data represented as mean ± SEM; statistical significance determined by unpaired two-sided $t$-

test). **j** Immunoblotting for indicated protein expression in PC-3 cells treated with different doses of ORY-1001 for 24 h ($n = 3$ independent experiments). **k**, **l** Colony formation assay (**k**) and transwell migration assay (**l**) for FOXA2-overexpressing PC-3 cells treated with either vehicle or ORY-1001 (10 μM, 10 d for colony formation assay, 2d for migration assay). **m**, **n** PC-3 stable cells (shNTC versus shFOXA2) (**m**) or parental cells (**n**) were subcutaneously injected into male mice. Mice bearing parental tumors were then treated with the LSD1 inhibitor ORY-1001 (0.03 mg/kg, daily intraperitoneal injection) (**n**). Tumor growth was measured at indicated time points (FOXA2 silencing experiment, $n = 8$ independent tumors; ORY-1001 treatment experiment, $n = 9$ independent tumors; data represented as mean ± SEM; statistical significance determined by two-way ANOVA). **o**, **p** GFP-labeled PC-3 stable cells (shNTC versus shFOXA2) (**o**) or parental PC-3 cells treated with ORY-1001 (10 μM, 24 h) (**p**) were injected into zebrafish embryos. Tumor cell invasion was immediately examined within 1 h and images were taken under 50x magnification. Embryos exhibiting positive circulation signals were classified as "invaded". The number represents the proportion of "invaded" embryos relative to the total number of injected embryos. ns ($P > 0.05$), *(0.01 < $P$ < 0.05), **(0.001 < $P$ < 0.01), ***($P < 0.001$), and ****($P < 0.0001$) were used to indicate the levels of $P$-value. Source data are provided as a Source Data file.

## JUN promotes tumor growth of DNPC and NEPC models

We next examined the transcription program of JUN by performing RNA-seq analyses in PC-3 and NCI-H660 cells with JUN silencing. In PC-3 cells, JUN activated cell cycle, Aurora kinase and PLK1 pathways and repressed immune response pathways, while in NCI-H660 cells JUN activated cell polarity, endothelin and ARF6 pathways (Fig. 6a, b and Supplementary Fig. 7a–c). Interestingly, while FOXA2 chromatin binding was associated with both transcriptional activation and repression functions in PC-3 cells (Supplementary Fig. 7d), the chromatin binding of JUN was predominantly associated with transcriptional activation in this model (Fig. 6c). However, in the NCI-H660 model, it appears that JUN was associated with both activation and repression functions (Fig. 6d). Putative direct targets of JUN—identified using BETA from the PC-3 and NCI-H660 datasets—were markedly overexpressed in AR-low CRPC samples, despite *JUN* mRNA levels being similar in AR-high and AR-low tumors (Supplementary Fig. 7e, f). Moreover, these JUN targets were globally suppressed by FOXA2 silencing or LSD1 inhibition (Fig. 6e, f), indicating that the transcriptional activity of JUN is dependent on unmethylated FOXA2. This result was confirmed by qRT-PCR analysis of a subset of FOXA2/JUN target genes (Fig. 6g).

We then developed a 40-gene FOXA2/JUN co-target signature from our PC-3 datasets (Supplementary Fig. 7g). This signature was more significantly correlated with LSD1 expression in AR-low CRPC tumors compared to AR-high CRPC tumors (Supplementary Fig. 7h) and was also associated with worse survival of patients in response to ARSi treatment (Fig. 6h). These results suggest a potential role of JUN, in collaboration with FOXA2, in driving CRPC cells to an AR-independent, therapy-resistant state. While treatments targeting AP-1 pathway have been developed and tested in other cancer types, this strategy has not been evaluated in CRPC. To address this, we tested a small molecule inhibitor of AP-1, T5224, which functions to interfere with the DNA binding of JUN or FOS family proteins[38] and is currently in phase 2 trials of inflammatory diseases in Japan (Toyama Chemical). As shown in Fig. 6i, FOXA2-dependent PC-3 cells showed greater sensitivity to this inhibitor than FOXA1-dependent LNCaP cells. Importantly, in xenograft models, T5224 significantly reduced the growth of PC-3 and NCI-H660 xenograft tumors (Fig. 6j, k).

To further validate these findings, we also examined a Foxa2-expressing murine PCa model, DKO cells, derived from the murine *Pten[-/-]/Rb[-/-]* transgenic model[10] (Supplementary Fig. 8a). The transcriptome of Foxa2 in this model is highly enriched for AP-1 signaling and stem cell-like lineage signatures (Supplementary Fig. 8b–e). Subsequently, we performed ChIP-seq analysis of Foxa2 in DKO cells. We found that Foxa2 chromatin binding sites (5,715 peaks) were

significantly associated with its transcriptional activation function and enriched for FOXA and AP-1 protein binding motifs (Supplementary Fig. 8f, g). Moreover, Lsd1 inhibition led to a global decrease in Foxa2 chromatin binding (Supplementary Fig. 8h), an effect that was further confirmed at two specific Foxa2 binding sites (Supplementary Fig. 8i). We also detected strong Jun binding at these sites, which was further decreased by silencing Foxa2 (Supplementary Fig. 8j), consistent with findings in human PCa lines. Finally, we demonstrate that DKO cells were more sensitive to LSD1 or AP-1 inhibitor treatments than SKO cells (*Pten[-/-]* only) (Supplementary Fig. 8k, l). Collectively, these findings indicate that AP-1 is a key driver and a potential therapeutic target in AR-independent CRPC.

## JUN regulates lineage-specific super-enhancers

Previous studies have shown that tumor progression is linked to the super-enhancer (SE) driven transcriptional network[39] and our recent results suggest an important function for LSD1 in forming liquid-liquid phase separation and activating CRPC-specific SEs[40]. Therefore, we assessed whether FOXA2/JUN can also regulate SEs in PC-3 and NCI-H660 cells. Using the ROSE algorithm[41,42] on global H3K27ac and/or H3K4me2 ChIP-seq data[43], we identified SEs in PC-3 and NCI-H660 cells (Supplementary Fig. 9a–c). FOXA2, JUN, and FOSL1 (only for PC-3) exhibited a strong enrichment at SEs compared to typical enhancers; this enrichment was equivalent to LSD1 and BRD4, factors that are known to play a key role in regulating SEs[41,44,45] (Fig. 7a and Supplementary Fig. 9d). Importantly, FOXA2 silencing caused a significant decrease of JUN/FOSL1 binding at SEs (Fig. 7b and Supplementary Fig. 9e). By comparing SEs in the LNCaP model (Supplementary Fig. 9f, g) with those in PC-3 and NCI-H660, we identified SEs that were unique to each model (305 for PC-3, 412 for NCI-H660, and 67 for LNCaP) (Supplementary Fig. 10a). Demonstrating the functional relevance of these unique SEs, genes associated with them were enriched in different pathways and distinctly overexpressed in each model (Supplementary Fig. 10b, c). For example, PC-3-specific SEs are enriched for cell-matrix adhesion and migration pathways while NCI-H660-specific SEs are enriched for cell fate and neuroendocrine development pathways.

We next examined whether FOXA1 or FOXA2 are enriched in lineage-specific SEs that are present in 201.1/2 PDX models. ChIP-seq analyses of H3K27ac were performed from 201.1 and 201.2 tumors and the ROSE algorithm was used to identify SEs (Supplementary Fig. 11a, b). Most of the identified SEs were unique to each model (724 for 201.1, 394 for 201.2) (Supplementary Fig. 11c). Importantly, FOXA1 and FOXA2 were specifically enriched at the 201.1 and 201.2 SEs, respectively (Supplementary Fig. 11d), suggesting that FOXA factors

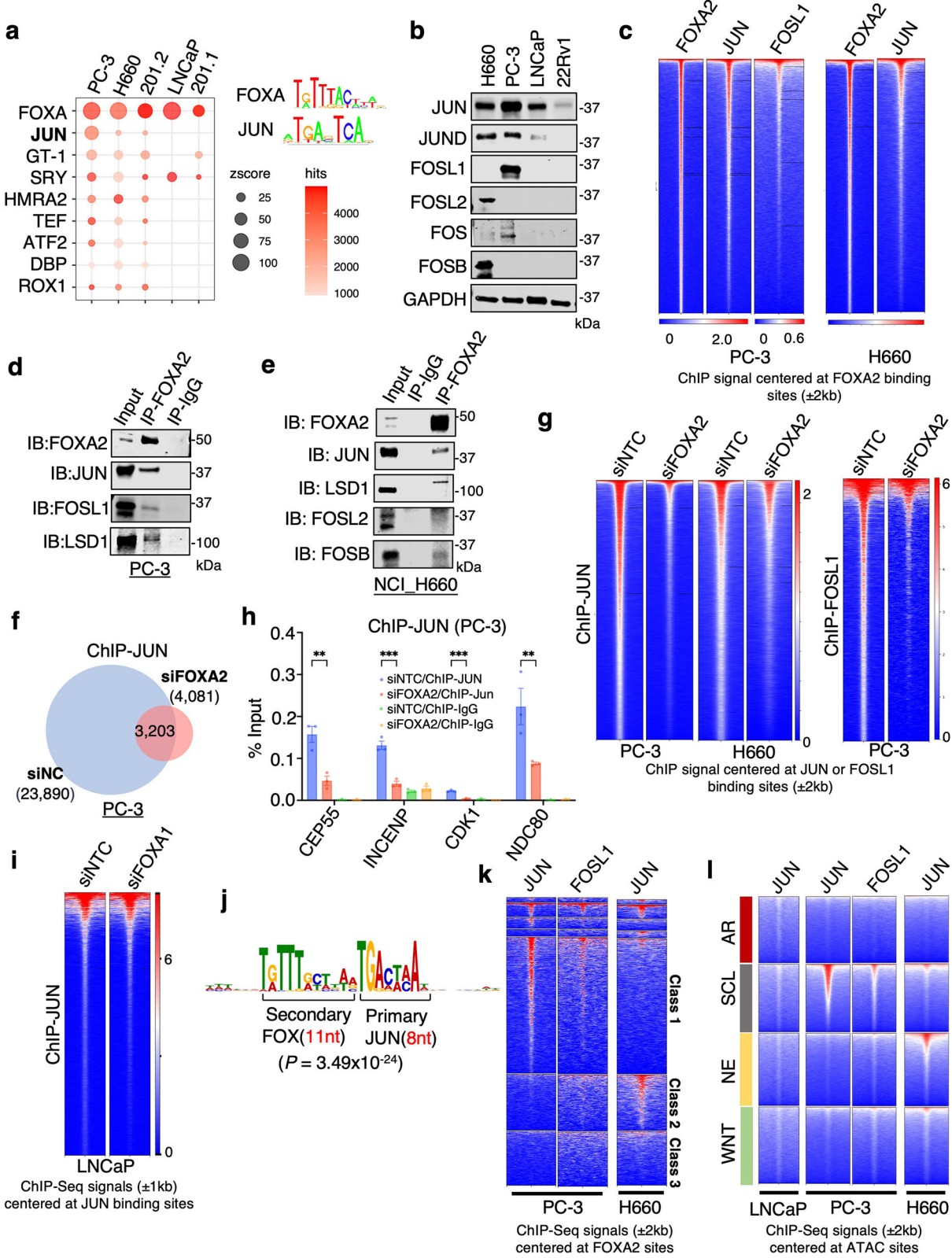

are important for regulating lineage-specific SEs. Genes associated with 201.1-specific SEs enrich for known AR function, such as regulation of fatty acid metabolism, while genes associated with 201.2-specific SEs enrich for WNT signaling and neuronal function (Supplementary Fig. 11e). To evaluate the function of SEs identified in mediating distinct CRPC lineages, we generated SE-associated gene signatures (Fig. 7c) and assessed their overlap with tumor molecular

subtypes. This analysis demonstrated that SEs are strongly associated with CRPC molecular subtypes in the SU2C mCRPC dataset, suggesting that they play a critical role in mediating PCa lineage progression (Fig. 7d).

Moreover, the transcript levels of genes associated with cell type-specific SEs were globally decreased by depletion of JUN (Fig. 7e), indicating a broader function of JUN in activating lineage-specific SEs.

**Fig. 5 | FOXA2 functions as a major pioneer factor of JUN. a** Enriched motifs identified by the SeqPos motif tool for FOXA2 binding sites in PCa models. **b** Immunoblotting for indicated protein in different PCa cell lines (*n* = 3 independent experiments). **c** Heatmap view for FOXA2, JUN, or FOSL1 ChIP-seq signal intensity centered at FOXA2 binding sites in PC3 or NCI-H660 cells. **d**, **e** Immunoblotting for indicated proteins that were coimmunoprecipitated with FOXA2 in PC3 (**d**) or NCI-H660 cells (**e**) (*n* = 3 independent experiments). **f** Venn diagram for ChIP-JUN peaks in PC-3 cells transfected with siFOXA2 versus siNTC. **g** Heatmap view for JUN or FOSL1 ChIP-seq signal intensity at JUN or FOSL1 binding sites in PC-3 or NCI-H660 cells transfect with siFOXA2 versus siNTC. **h** ChIP-qPCR for JUN binding at the indicated FOXA2/JUN co-target sites (*n* = 3 independent

samples; data represented as mean ± SEM; statistical significance determined by unpaired two-sided *t*-test). **i** heatmap view for JUN ChIP-seq signal intensity at JUN binding sites in LNCaP cells transfected with siFOXA1 versus siNTC. **j** Spaced motif analysis using SpaMo was conducted to analyze the composition motif enrichment at the overlapping sites of FOXA2 and JUN in PC-3 cells. **k**, **l** Heatmap view for JUN or FOSL1 binding peak intensity centered at previously defined subclasses of FOXA2 binding sites (defined in Fig. 2c) (**k**), or at previously defined chromatin sites with different ATAC signatures (**l**) in LNCaP, PC3, or NCI-H660 cells. ns (*P* > 0.05), *(0.01 < *P* < 0.05), **(0.001 < *P* < 0.01), ***(*P* < 0.001), and ****(*P* < 0.0001) were used to indicate the levels of *P*-value. Source data are provided as a Source Data file.

One of the PC-3-unique SEs is located upstream of the *FOSL1* gene (Fig. 7f) and silencing FOXA2 significantly decreased JUN binding at this site (Supplementary Fig. 11f). Demonstrating the relevance of this SE, silencing either FOXA2 or JUN dramatically suppressed *FOSL1* mRNA and protein expression in PC-3 cells (Fig. 7g–i). A similar effect was also observed on another known FOXA2 target, *PTHLH*[46]. These results suggest a positive feedback mechanism by FOXA2 and JUN to strengthen AP-1 activity in the stem-cell like CRPC by robustly activating *FOSL1*. Altogether, these studies reinforce the notion that a FOXA1 to FOXA2 functional switch, in cooperation with JUN, can mediate CRPC lineage plasticity possibly via activating lineage-specific SEs.

### FOXA2 expression in AR-dependent PCa cells initiates a multilineage transition

Through analysis of H3K27ac ChIP-seq data in ADT-resistant LNCaP-abl cells[47] and parental LNCaP cells[26], we found that AR-targeting treatments can induce a multilineage transition state, accompanied by a gradual increase in FOXA2 expression[48] (Supplementary Fig. 12a, b). Given our data thus far, which shows that FOXA2 can drive distinct programs in promoting different lineages, we hypothesize that FOXA2 overexpression may play a key role in initiating this multilineage transition in AR-dependent luminal PCa tumors. To test this hypothesis, we stably overexpressed FOXA2 in FOXA2-negative LNCaP cells (LN-FOXA2-OE). Overexpression of FOXA2 (in hormone-depleted conditions) increased the expression of a canonical NE marker, ENO2, but not other widely used markers, such as CHGA (Fig. 8a). Interestingly, FOXA2 expression also led to a marked decrease in FOXA1 levels. This reduction in FOXA1 expression is likely an adaptive effect, as rapid silencing of FOXA2 in NCI-H660 cells did not significantly increase FOXA1 levels (Fig. 8b). Furthermore, FOXA2 overexpression enhanced the expression of identified FOXA2 targets and increased the proliferation and migration of LNCaP cells under castrated conditions (Fig. 8c–e).

To further evaluate the transcriptional role of FOXA2 in this model, we performed RNA-seq to define the FOXA2 transcriptome in LN-FOXA2-OE cells. As shown in Fig. 8f, FOXA2 activated cell cycle signaling (E2F and MYC targets) and Aurora kinase pathway as well as NOTCH and WNT signaling in LNCaP cells. AR signaling was modestly suppressed in LN-FOXA2-OE cells compared to parental cells (Supplementary Fig. 13a). Using the subtype-specific transcriptional signatures developed by Tang et al.[33], we observed that FOXA2 overexpression significantly activated NE and WNT transcription programs (Fig. 8g). Consistently, by using ChIP-seq to examine FOXA2 and JUN chromatin binding in these cells, we found that FOXA2 binding sites were enriched in all three non-AR classes whereas JUN binding sites were redistributed to these three classes but with stronger association with NE and WNT signature sites (Fig. 8h). Considering the potential demethylation of FOXA2 at the K265 site, we examined how the K265R mutation impacts JUN binding and lineage reprogramming. Notably, the overexpression of K265R mutant did not suppress FOXA1 expression compared to the WT (Supplementary Fig. 13b). However, this FOXA2 mutation expanded and strengthened JUN chromatin

binding, and further enhanced the multilineage redistribution of JUN binding sites (Supplementary Fig. 13c-f). Subsequent RNA-seq analysis in cells overexpressing K265R revealed that this mutation broadly enhanced transcriptional signatures in NE and WNT subtypes, compared to the WT FOXA2 expression (Supplementary Fig. 13g). Collectively, these findings indicate that FOXA2 overexpression triggers a multilineage transition of CRPC cells that is associated with reprogramming of JUN chromatin binding and regulated by its K265 methylation status.

### FOXA2 transcription programs are activated in a subset of AR-dependent CRPC tumors

Our established lineage-specific FOXA1/2 target signatures, which were specific to the models from which they were derived (Fig. 9a), were also altered in LN-FOXA2-OE cells. More specifically, all of the FOXA2 signatures—which were derived from three AR-independent models—were co-enriched in LN-FOXA2-OE cells and decreased by silencing JUN (Fig. 9b-c). Therefore, we next examined the expression of these FOXA1/2 signatures in the SU2C mCRPC dataset, which can be divided into 5 clusters based on transcriptomic data. As shown in Fig. 9d, FOXA1 targets identified from LNCaP/22Rv1[40] or 201.1 models were overexpressed in clusters 1 and 2, consistent with high AR scores in these subsets. FOXA2 targets from PC-3 cells were mainly overexpressed in clusters 4 and 5 but also have a major overlap with cluster 1. FOXA2 targets from NCI-H660 cells and 201.2 were overexpressed in clusters 3, 4, and 5 and have weak overlaps with clusters 1 and 2. The expression of multiple FOXA2 target signatures in CRPC-AR samples, primarily cluster 1, may indicate tumor subclones expressing FOXA2 and exhibiting early lineage reprogramming or may be due to FOXA2-initiated multilineage reprogramming in AR-dependent CRPC cells. Overall, these data strongly argue that the overexpression of FOXA2 can act as an early driver to induce multilineage reprogramming of CRPC cells in response to ADT or ARSi and highlight a phenomenon whereby switching from FOXA1 to FOXA2 pioneer factor activity causes loss of a CRPC-AR phenotype and enhances cancer cell plasticity, leading to the emergence of multiple new CRPC lineages (Fig. 9e).

## Discussion

Mounting evidence supports the notion that lineage plasticity is one of the major mechanisms by which prostate cancer tumors escape from ADT and ARSi[6,10,11,25,49–54]. While the neuroendocrine transformation, which accounts for 10–20% of overall CRPC cases, has been extensively studied due to its distinct cytologic change of tumor cells, the mechanisms driving other molecular subtypes of lineage reprogramming are poorly understood. DNPC, characterized by being AR-negative or AR-low and lacking expression of NE markers such as SYP or CHGA, represents another major molecular subtype of AR-independent CRPC and is regulated by distinct mechanisms[18,33,52,55]. DNPC is estimated to comprise up to 20–30% of overall CRPC cases[13,33]. Recent findings by Tang et al., suggest that DNPC can be further classified into SCL and WNT molecular subtypes based on unique chromatin accessibility signatures[33]. That study revealed that the SCL subclass is enriched for JUN/FOS complex activity, as determined by

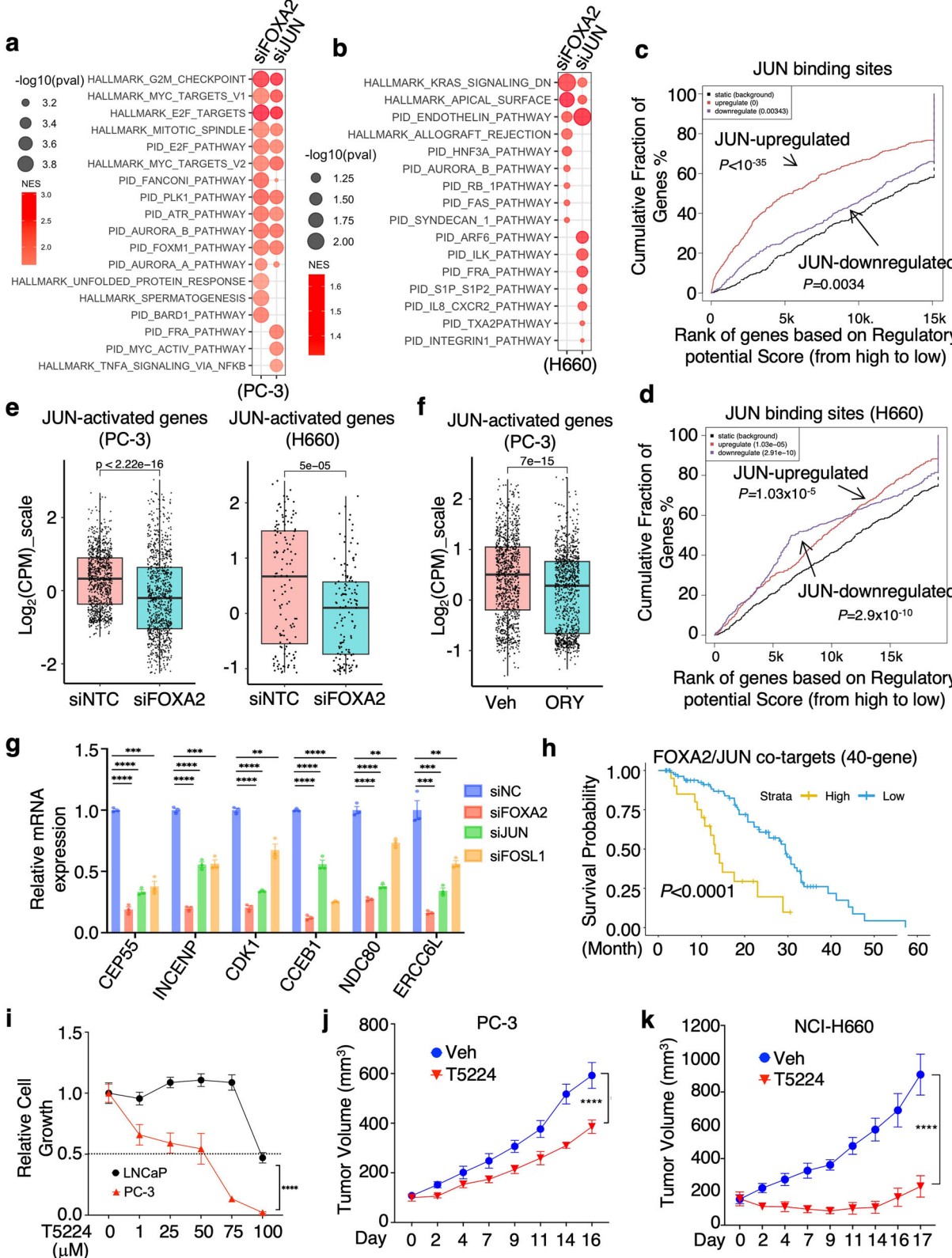

motif enrichment, while the WNT subclass is enriched for TCF/LEF activities. However, the epigenetic and transcriptional drivers for DNPC and NEPC remain to be determined.

FOXA family proteins are well-known pioneer factors that displace linker histones on nucleosomes, thereby opening the compact chromatin structure and increasing the accessibility of other transcription

factors[14]. While the function and regulation of FOXA1 in adenocarcinoma PCa have been extensively studied, the activity of FOXA2 has been poorly understood. A recent study using CRPC patient samples indicated that FOXA2 is overexpressed in the majority of NEPC samples, suggesting a distinct role of FOXA2 in NEPC development[23]. Interestingly, the study also found that ~5% of adenocarcinoma CRPC

**Fig. 6 | JUN promotes tumor growth of FOXA2-driven PCa models. a, b** GSEA for differentially regulated genes in PC-3 (**a**) or NCI-H660 (**b**) transfected with siNTC versus siFOXA2 or siJUN (RNA-seq data). The red color indicates enriched pathways for downregulated genes by FOXA2 or JUN silencing. **c, d** BETA for the association of JUN binding sites with JUN-regulated genes in PC-3 (**c**) or NCI-H660 (**d**) cells. **e, f** Boxplot view for the expression (Log$_2$(CPM_zscore)) of JUN-activated genes ($n = 1036$ genes for PC-3, n = 132 genes for NCI-H660) in PC-3 or NCI-H660 cells transfected with siNTC versus siFOX2 (**e**), or treated with/out ORY-1001 (10 μM for 24 h) (**f**) (center: median; box: 25th to 75th IQR; whiskers: 1.5x IQR; outliers: individual data points; statistical significance determined by unpaired two-sided $t$-test). **g** qRT-PCR for the expression levels of indicated FOXA2-JUN cotargets in PC-3 cells treated with siFOXA2, siJUN, siFOSL1, or siNTC ($n = 3$ independent samples; data represented as mean ± SEM, statistical significance determined by unpaired two-sided $t$-test). **h** Kaplan–Meier curve for the overall survival from the start of a first-line ARSi in CRPC tumors (SU2C dataset, $n = 106$) with higher FOXA2-JUN co-target signature (red, the top 25%) versus lower (blue, the bottom 75%). **i** Cell viability assay for LNCaP and PC-3 cells treated with 0–100 μM T5224, an AP-1 inhibitor, for 3d (LNCaP, $n = 5$ independent samples; PC-3, $n = 4$ independent samples; data represented as mean ± SEM; statistical significance determined by two-way ANOVA). **j, k** PC-3 (**j**) or NCI-H660 (**k**) cells were subcutaneously injected into male mice. Mice bearing parental tumors were then treated with T5224 (6 mg/kg, 5 days per week via gavage). Tumor growth was measured at indicated time points (PC-3, $n = 12$ independent tumors; NCI-H660, n = 8 independent tumors; data represented as mean ± SEM; statistical significance determined by two-way ANOVA). ns ($P > 0.05$), *($0.01 < P < 0.05$), **($0.001 < P < 0.01$), ***($P < 0.001$), and ****($P < 0.0001$) were used to indicate the levels of $P$-value. Source data are provided as a Source Data file.

cases express high levels of FOXA2. These tumors may represent other FOXA2-driven CRPC subtypes. Using a transgenic mouse model with $Pten^{-/-}/Rb^{-/-}/Tp53^{-/-}$ background, Han et al., reported that an increase in *FOXA2* expression, along with a decrease in *FOXA1*, can drive neuroendocrine transformation from luminal PCa cells[25]. However, the molecular activity of FOXA2 has not been well-defined, and its cooperative transcription factors remain unknown. In this study, we investigated the function of FOXA2 using a variety of models collectively including multiple AR-independent phenotypes: PC-3, a DNPC line classified as an SCL subtype; NCI-H660, a NEPC line; and a pair of PDX models: FOXA1-high PDX-201.1A-Cx and FOXA2-high PDX-201.2A-Cx, which were established from different metastatic sites of the same patient[30]. Our analysis using ChIP-H3K27ac further determined that 201.2 has a unique WNT signature of chromatin accessibility. Importantly, the transcriptomes of PC-3, NCI-H660, and 201.2 were all globally associated with FOXA2 activity, demonstrating its role as a pan-plasticity factor that can drive the emergence of multiple AR-independent CRPC lineages.

Our findings clearly demonstrate that JUN is a major cooperative transcription factor of FOXA2, but not FOXA1, as JUN chromatin binding is not enriched in FOXA1-occupied enhancers in ARPC models. The differential chromatin binding of FOXA2 redistributes JUN to lineage-specific enhancers, including super-enhancers, leading to the transcriptional reprogramming of AP-1. Interestingly, while JUN expression is generally constant across all subtypes of CRPC, FOS family proteins may exhibit context-dependent expression—for example, FOSL1 is overexpressed in the SCL subtype. Therefore, different AP-1 complexes may function in different CRPC subtypes to reprogram their transcription networks. This finding is consistent with a recent report on pancreatic tumors, where FOXA2 can drive tumor heterogeneity by interacting with different cooperative factors, including AP-1[56]. Additionally, we also revealed a positive feedback mechanism mediated by FOXA2 and JUN that can activate the super-enhancer of *FOSL1* and suggest that FOSL1 could be a biomarker for stem-cell-like DNPC, which is also consistent with a previous report[33]. Thus, our study indicates that FOXA2 may have a broader function in driving lineage plasticity in other subtypes in addition to NEPC. While FOXA2 clearly drives the reprogramming of JUN, its expression alone in luminal ARPC cells does not determine the fate of lineage reprogramming. Instead, it appears that FOXA2 primes tumor cells for a multilineage phase. Indeed, overexpressed FOXA2 binds to all three subclasses of enhancers that can drive SCL, NE, and WNT subtypes, and also facilitates the binding of JUN to all three classes of enhancers, although JUN binding is more associated with WNT and NE signature sites as seen in the LNCaP model. These data are intriguing, as they suggest that additional mechanisms may further restrict FOXA2/AP-1 binding, potentially determining the final state of lineage transition. Since DNPC shares most genetic alterations with NEPC, it is possible that this lineage determination mechanism is dependent on later epigenetic signals and tumor microenvironment.

Our previous studies indicated a tight regulation of FOXA1 activity by LSD1 in ARPC models though LSD1-mediated demethylation of FOXA1 protein[26,29]. In this study, we found that LSD1 can similarly stabilize FOXA2 chromatin binding via demethylation of K265. However, it remains unclear whether LSD1 is directly involved in determining lineage reprogramming, as we observed that completely demethylated FOXA2 is incapable of suppressing FOXA1 expression and may thus maintain the AR program in CRPC cells. This phenomenon suggests that the methylation/demethylation status of FOXA2 needs precise regulation and balance during lineage reprogramming. Interestingly, we recently identified SETD7 as the methyltransferase of FOXA1 and potentially FOXA2 as well and demonstrated that SETD7 can function as a gate-keeper at many FOXA motif-containing chromatin regions to prevent unwanted FOXA protein binding[57]. Hence, the combined activities of FOXA2 methylation/demethylation could further define its chromatin binding patterns and its suppressive activity on FOXA1. Future studies are needed to validate this hypothesis and precisely define the mechanisms by which the combined actions of LSD1/SETD7 influence the pro-plasticity activity of FOXA2.

Importantly, our study suggests two therapeutic strategies to target the FOXA2-AP-1 pathway. One is to use an LSD1 inhibitor, which is currently being extensively tested in clinical trials. Our recent preclinical studies using a series of CRPC-AR models have demonstrated the efficacy of LSD1 inhibitors alone or in combination with BRD4 inhibitors[26,40]. Interestingly, the efficacy of these treatments in these models appears to be dependent on FOXA1 levels. Results from the current study suggest that LSD1 inhibitors could be even more effective in DNPC and NEPC models by targeting FOXA2 chromatin binding. The second therapeutic strategy is to target AP-1 activity. In this study, we demonstrated the efficacy and low toxicity of a small molecule inhibitor that blocks FOS/JUN chromatin binding in both DNPC and NEPC xenograft models. Another rationale for targeting LSD1 and JUN is that they are both critical regulators of lineage-specific super-enhancers. We recently reported that FOXA1 can recruit BRD4 and may form nuclear condensates with LSD1 and BRD4 to regulate super-enhancers[40]. In this study, we demonstrate that FOXA2 and AP-1 can also regulate the lineage-specific super-enhancers in AR-null models, potentially via a mechanism involving LSD1 and BRD4. Therefore, combinatorial treatments comprising JUN inhibitors, LSD1 inhibitors, and potentially BET inhibitors could be a rational strategy to tackle lineage-specific oncogenic super-enhancers. Since these inhibitors are all under clinical trials, preclinical studies to demonstrate the efficacy of these treatments are urgently needed, and findings from those studies may be rapidly translated into clinical testing of AR-independent CRPC.

In summary, our study defines FOXA2 as an early driver of the PCa lineage plasticity by redistributing JUN chromatin binding to lineage-specific developmental enhancers, thereby inducing the transcriptional reprogramming of AP-1. This master regulator network drives reprogramming to multiple AR-independent lineages in CRPC through initiating a multilineage transition state. Future

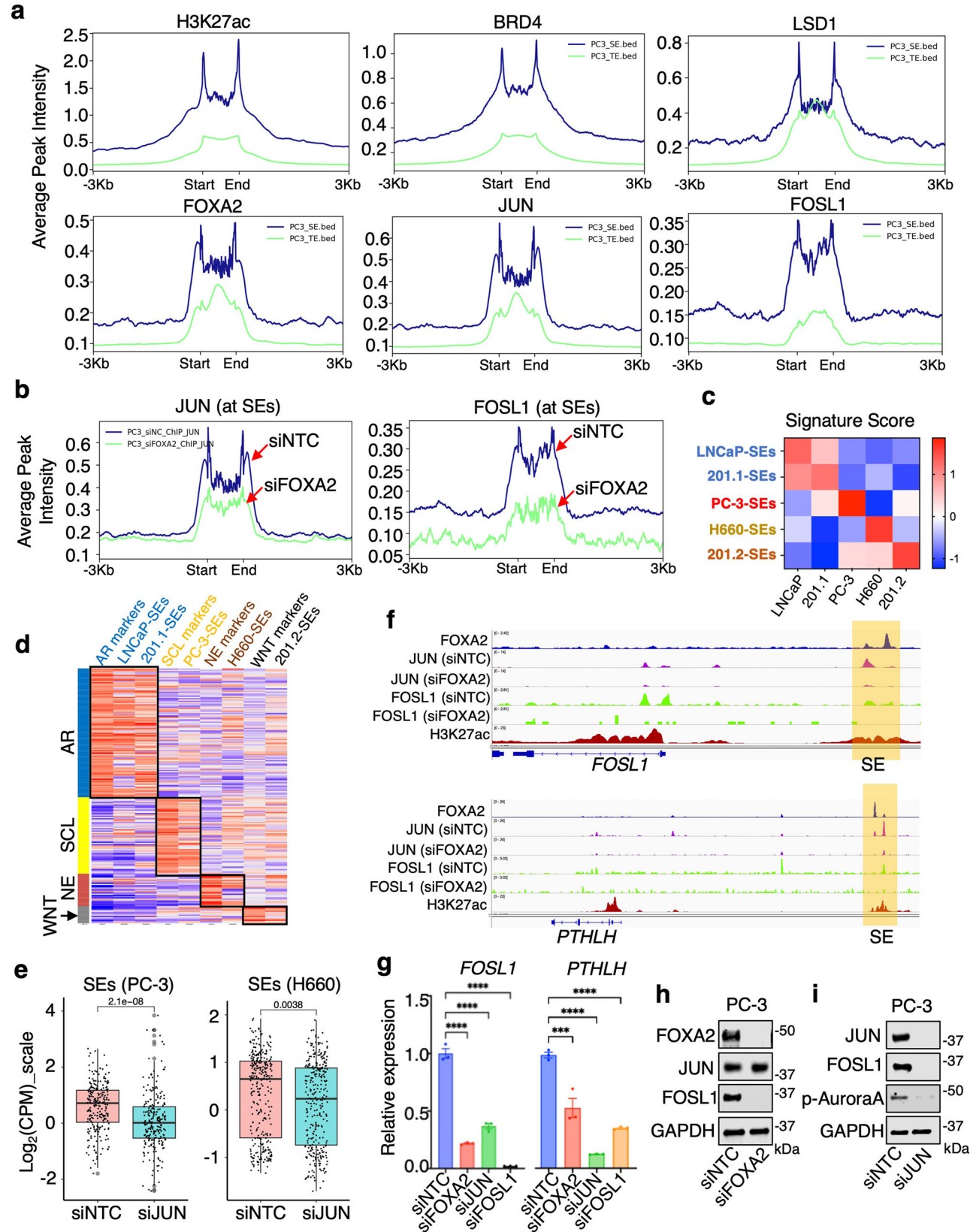

studies should focus on using single-cell omics approaches in patient samples with a range of genomic backgrounds to comprehensively dissect the plasticity-promoting activity of FOXA2, LSD1, and JUN. This will provide further insights into the mechanism underlying lineage plasticity and may help develop therapies for AR-independent CRPC.

## Methods

### Cell culture, plasmids, stable cell line generation, and transfection

PC-3, LNCaP, CWR-22Rv1, NCI-H660, and 293 T cells were purchased from ATCC and authenticated periodically using short tandem repeat (STR) profiling. SKO and DKO murine prostate cancer cell lines were

**Fig. 7 | JUN regulates lineage-specific super-enhancers. a** The average binding intensity of indicated proteins at super-enhancers (SEs) versus typical enhancers (TEs) in PC-3 cells. **b** JUN and FOSL1 binding intensity at SEs sites in PC-3 cells transfected with siFOXA2 versus siNTC. **c, d** Heatmap view for GSVA (gene set variation analysis) scores of genes associated with top 50 SEs identified from each model in PCa cells (**c**) or in SU2C mCRPC cohorts (**d**). **e** Boxplot view for the expression (Log$_2$(CPM_zscore)) of model-specific SE-associated genes in PC-3 ($n = 305$ genes) or NCI-H660 ($n = 412$ genes) cells transfected with siJUN versus siNTC (center: median; box: 25th to 75th IQR; whiskers: 1.5x IQR; outliers: individual data points; statistical significance determined by unpaired two-sided *t*-test).

**f** Genome browser view for indicated ChIP-seq peaks (from the PC-3 model) at *FOSL1* and *PTHLH* gene locus. **g** qRT-PCR for the expression of *FOSL1* and *PTHLH* in PC-3 cells transfected with siFOXA2, siJUN, siFOSL1, or siNTC ($n = 3$ independent samples; data represented as mean ± SEM; statistical significance determined by unpaired two-sided *t*-test). **h, i** Immunoblotting for indicated proteins in PC-3 cells transfected with siNTC versus siFOXA2 (**h**) or siJUN (**i**) ($n = 3$ independent experiments). ns ($P > 0.05$), *($0.01 < P < 0.05$), **($0.001 < P < 0.01$), ***($P < 0.001$), and ****($P < 0.0001$) were used to indicate the levels of *P*-value. Source data are provided as a Source Data file.

provided as gifts from Dr. Leigh Ellis. PC-3, LNCaP, and CWR-22Rv1 cells were cultured in RPMI-1640 supplemented with 10% fetal bovine serum (FBS). NCI-H660 cells were cultured in RPMI-1640 base medium supplemented with 5% FBS, 10 nM β-estradiol, 10 nM hydrocortisone, and 1% Insulin-Transferrin-Selenium. 293 T, SKO (*Pten$^{-/-}$*), and DKO (*Pten$^{-/-}$/Rb$^{-/-}$*) murine PCa cell lines were cultured in DMEM supplemented with 10% FBS. We routinely test these lines for Mycoplasma contamination using the MycoAlert Mycoplasma Detection Kit by Lonza. siRNAs (ON-TARGETplus) and shRNA lentiviral particles (pGIPZ) were pre-designed and acquired from Dharmacon. The 3xFLAG-FOXA2$^{WT}$ plasmid (EX-Y4558-Lv181) and the control plasmid (EX-NEG-Lv181) were purchased from GeneCopoeia. The 3xFLAG-FOXA2$^{WT}$ plasmid was then used as the template to generate 3xFLAG-FOXA2$^{K265R}$ (AAG- > CGG) using the QuickChange Lightning Site-Directed Mutagenesis Kit (Agilent Technologies) per the manufacturer's protocol. RNAimax transfection reagent (Thermo Fisher) was used for transfection, and viruses were packaged in 293 T cells. To establish stable cell lines, lentivirus was used to infect PC-3 and LNCaP cells, which were then selected with puromycin. For the cell treatment with LSD1/AP-1 inhibitors, we initially prepared a high-concentration stock solution in DMSO. This was then diluted in the culture medium to achieve the desired concentration. For the control treatment, an equivalent volume of DMSO, as used in the inhibitor solution, was utilized.

### Patient-derived Xenografts
PDX 201.1A-Cx and PDX 201.2A-Cx were previously generated by Melbourne Urological Research Alliance (MURAL)[30,31,58]. The samples were collected through the CASCADE rapid autopsy program with informed written consent. The studies were conducted under Human Research Ethics Committee (Institutional Review Board) approvals at Monash University (7996, 12287) and the Peter MacCallum Cancer Centre (15/98, 97_27). The PDXs were grown in male NSG mice according to animal ethics approvals at Monash University (MARP 2014/085, 28911) and regularly authenticated using STR profiling. Immunohistochemistry was used to confirm the phenotype of the PDXs and the lack of lymphoma. Bulk RNA-sequencing data (Lexogen 3′ Quantseq FWD + Illumina HiSeq 2000/2500) for PDXs was previously reported with Xenomapper (1.0.1) used to select human versus mouse reads[31].

### Cell viability, cell cycle, migration, invasion, and colony formation assays
The viability of the cells was assessed by utilizing the CellTiter-Glo Luminescent Cell Viability Assay (Promega, USA). Cell cycle assay was performed using Muse® Cell Cycle kit (Thermo Fisher). The migration assay was conducted using Corning FluoroBlok™ cell culture inserts (Falcon, 351152). The invasion assay was conducted using Corning® BioCoat™ Matrigel® Invasion Chambers (Corning, 354480) following the manufacturer's protocol. To perform the colony formation assay, each well of 6-well plates were seeded with 1000 cells per well. After 10 day incubation, cells were fixed and stained using Crystal Violet (Fisher, C581-25) to visualize and count the colonies.

### Total RNA extraction, qRT-PCR, and RNA-Seq analysis
TRIzol reagent (Invitrogen) or RNeasy Kit (Qiagen) were used for RNA extraction. qRT-PCR was performed using Taqman probe/primer mixes and Fast 1-step Mix (Thermo Fisher Scientific) on QuantStudio 3 PCR machine. The qRT-PCR results were normalized with GAPDH and quantified using the ΔΔCt method. The TaqMan primers and probes for *KDM1A*, *FOXA2*, *FOSL1*, *CEP55*, *NDC80*, *INCENP*, *CDK1*, *TP63*, *CCEB1*, *PTHLH*, *ERCC6L* and *GAPDH* were purchased as pre-designed mixes from Thermo Fisher Scientific.

For RNA-seq, The TruSeq Strand Total RNA LT kit (Illumina) was used for library preparation. Illumina HiSeq2500 (51 nt, single-end) or Illumina NextSeq 2000 (51nt, pair-end) were used for next-generation sequencing. The human reference genome (hg19) was used to align transcriptome-sequencing reads by using STAR (version 2.7.1a). featureCounts (version 2.0.1) from GRCh37 Ensembl reference was used for counting. R package Edger (3.36.0) was then employed to process all gene counts and evaluate the differential expression by using the Benjamini-Hochberg false discovery rate (FDR)-adjusted *P*-value. The expression values were normalized by centering and scaling across samples and displayed using the ComplexHeatmap (version 2.10.0) R package. Gene Set Enrichment Analysis (GSEA) was performed using Software GSEA (version 4.2.2) and R package fgsea (version 1.20.0).

### Immunoblotting
For the extraction of proteins, cells were lysed using 2% SDS and boiled for 5 minutes (mins). For tissue protein isolation, tissue samples were bead-milled (5 mm) in RIPA buffer containing protease inhibitors using the TissueLyser LT (Qiagen). Protein samples were loaded onto 4-15% Mini-PROTEAN TGX precast protein gels (Bio-Rad) and transferred onto nitrocellulose membranes (Bio-Rad). The membranes were then blocked with 5% non-fat milk and incubated overnight at 4 °C with the following antibodies: anti-LSD1 (Abcam, 1:1000), anti-FOXA2 (Millipore,1:1000), anti-FOXA2 (Proteintech,1:1000), anti-JUN (CST,1:1000), anti-Aurora A (CST,1:1000), anti-p-Aurora A (CST,1:1000), anti-H3K4me2 (Millipore, 1:1000), anti-Methyl-Lysine (Abcam, 1:200), anti-CHGA (Abcam,1:1000), anti-Ncad (Abcam,1:1000), anti-Ecad (Sigma,1:1000), anti-FLAG (Sigma,1:1000), anti-FOSL1 (Abcam,1:1000), anti-FOSL2 (Abcam,1:1000), anti-FOSB (Abcam,1:1000), anti-FOS (Abcam,1:1000), anti-V5 (Sigma,1:1000), anti-FOXA1 (Abcam,1:1000), and anti-GAPDH (Abcam,1:5000). Membranes were subsequently incubated with secondary antibodies labeled with fluorescence (LI-COR Biosciences) in 5% non-fat milk for 1 hour (h) at room temperature. Gel images were captured using the LI-COR Odyssey system at a wavelength of 680 nm or 800 nm. Secondary antibodies used were goat anti-rabbit (LI-COR IRDye 800CW, 1:3000) and anti-mouse (LI-COR IRDye 680RD, 1:3000).

### Immunoprecipitation and mass spectrometry
To perform endogenous FOXA2 immunoprecipitation, cells were lysed using Co-IP lysis buffer containing Tris-HCL (20 mM), NaCl (150 mM), EDTA (5 mM), DTT (mM), and Triton X100 (0.5%), along with protease inhibitors (Thermo Fisher). The lysates were then pre-cleared using IgG-conjugated beads (Sigma) for 1 h at 4 °C. Equal amounts of protein

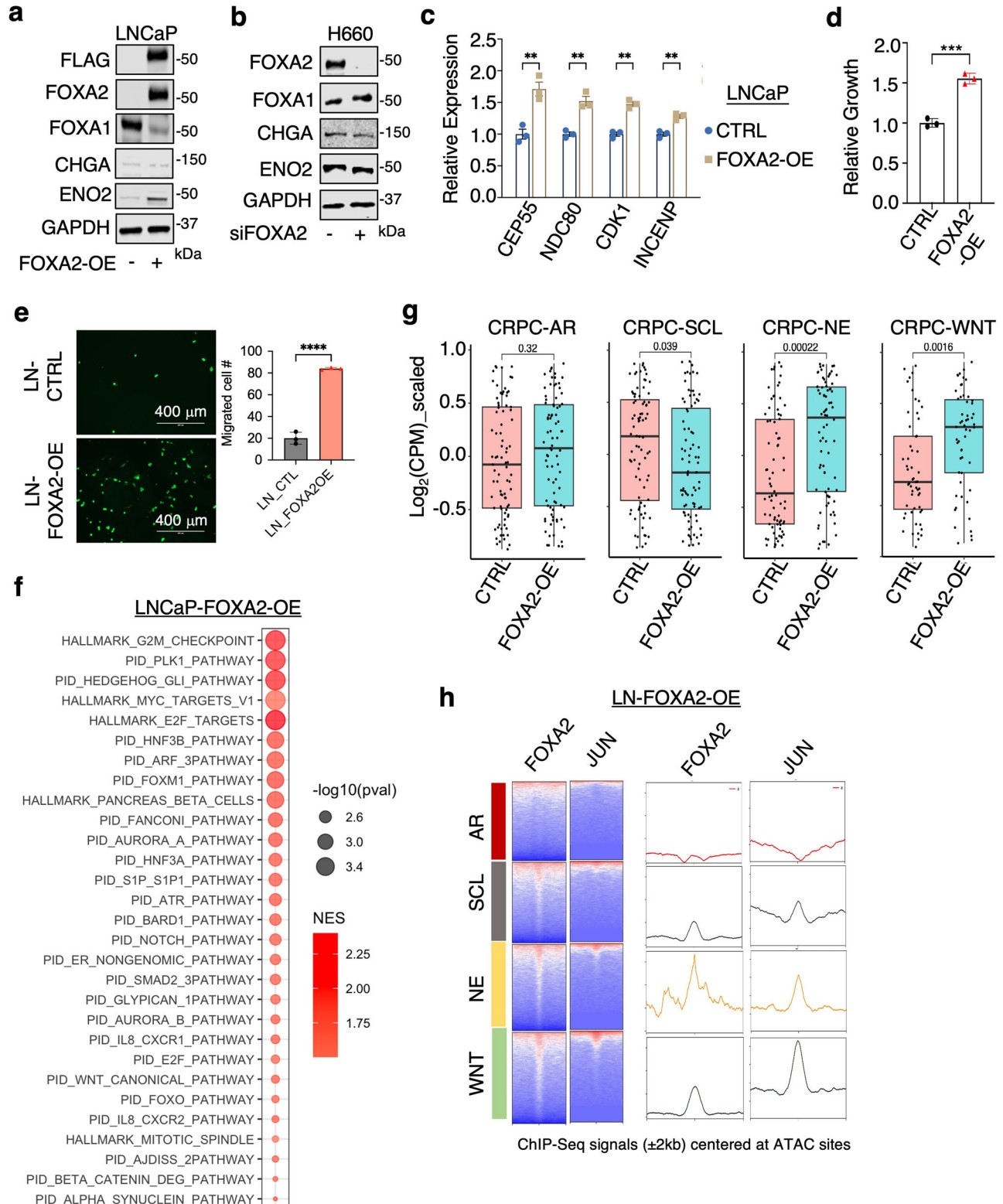

(1–5mg) were then incubated with FOXA2 antibody for overnight. For FLAG pull-down, protein extracts of cells stably expressing FLAG-tagged FOXA2 were incubated with 15 µl of FLAG-conjugated beads (Sigma, F2426) for overnight. To perform mass spectrometry analysis, we used at least $6 \times 10^8$ cells to map post-translational modification sites through Thermo EASY-nLC 1200 at the Proteomics Core of University of Massachusetts Boston.

**ATAC-seq**

We conducted Omni ATAC-seq as previously described[59]. Briefly, we collected ~50,000 viable PC-3 or NCI-H660 cells and centrifuged them at 4 °C. We used the Illumina Tagment DNA TDE1 Enzyme and Buffer Kit for library preparation, followed by immediately cleaning the DNA samples using the Qiagen QIAquick Purification Kit and pre-amplifying them with NEBNext High-Fidelity 2× PCR Master Mix.

**Fig. 8 | FOXA2 expression in AR-dependent PCa cells initiates a multilineage progression. a, b** Immunoblotting for indicated proteins in LNCaP cells stably expressing empty vector or FOXA2 (LN-FOXA2-OE) (**a**) or in NCI-H660 cells transfected with siFOXA2 versus siNTC (**b**) (*n* = 3 independent experiments). **c** qRT-PCR for indicated FOXA2 targets in the control LNCaP versus LN-FOXA2-OE cells (*n* = 3 independent samples, data are represented as mean ± SEM, statistical significance determined by unpaired two-sided *t*-test). **d, e** Cell proliferation assay (**d**) or transwell migration assay for the control LNCaP versus LN-FOXA2-OE cells (**e**) (*n* = 3 independent samples, data are represented as mean ± SEM, statistical significance determined by unpaired two-sided *t*-test). **f** GSEA for the pathways (HALLMARK and

PID datasets) enriched in the upregulated genes by FOXA2 overexpression. **g** Box plots for subtype-specific transcriptional signatures (*n* = 93 genes for every subtype) in the control LNCaP versus LN-FOXA2-OE cells (center: median; box: 25th–75th IQR; whiskers: 1.5x IQR; outliers: individual data points; statistical significance determined by unpaired two-sided *t*-test). **h** Heatmap view (left panel) and reads density plot (right panel) for FOXA2 or JUN binding peak intensity centered at previously defined chromatin sites with different ATAC signatures in LN-FOXA2-OE cells. ns (*P* > 0.05), *(0.01 < *P* < 0.05), **(0.001 < *P* < 0.01), ***(*P* < 0.001), and ****(*P* < 0.0001) were used to indicate the levels of *P*-value. Source data are provided as a Source Data file.

Additional cycles were determined using qPCR amplification to prevent over-amplification. The final PCR product was purified to remove primer dimer and large fragments using AMPure XP and assessed for quality control using the Agilent High Sensitivity Screen Tape. The libraries were sequenced on Illumina NextSeq 2000 genome analyzer.

### ChIP, re-ChIP, and ChIP-Seq analysis

To prepare the Chromatin Immunoprecipitation assay (ChIP), cells were fixed with 1% formaldehyde and lysed using ChIP lysis buffer (1% SDS, 5 mM EDTA, 50 mM Tris–HCl pH 8.1). Chromatin was then sheared to produce either ~500–800 bp fragments (for ChIP-qPCR) or ~300 bp fragments (for ChIP-seq) using a Bioruptor Sonicator (Diagenode). Immunoprecipitation was carried out using ChIP-grade antibodies: anti-H3K27ac (Abcam), anti-H3K4me2 (Abcam), anti-FOXA2 (Abcam), anti-JUN (CST), anti-FOSL1 (Abcam). Precipitated protein-DNA complexes were reverse-crosslinked at 65 °C, followed by DNA purification. The extracted DNA was then subjected to ChIP-qPCR using SYBR green or ChIP-seq analysis. For re-ChIP assay, to release the first-round ChIP-DNA, 100 ul of 10 mM DTT was incubated at 37 °C for 30 min, followed by two washes of the beads and collection of the DTT. A second ChIP-grade antibody was used for the second round of immunoprecipitation.

ChIP-seq libraries were constructed using the SMARTer ThruPLEX DNA-Seq Prep Kit (Takara Bio USA), and next-generation sequencing (51nt, pair-end) was performed using Illumina NextSeq 2000. The ChIP-seq (and ATAC-seq) reads were mapped to the hg19 human genome using bwa (version 0.7.17-r1188) with the aln and sampe subcommands. Samtools (version 1.9) was used to convert sam files to bam format. Enriched ChIP regions were evaluated using MACS2 (version 2.1.4). The Intervene (version 0.6.5) was used to analyze peak intervals, determine overlapped regions, and generate Venn diagrams. The signals associated with genomic regions were visualized using compueMatrix and plotHeatmap tools from deepTools (version 3.3.0). computeMatrix was used to calculate scores for each genomic region and plotHeatmap was used to create a heatmap for scores associated with genomic regions. Motif enrichment analysis was performed using SeqPos with default settings. Binding and Expression Target Analysis (BETA) was performed using the BETA software package (version 1.0.7).

### In vitro demethylation assay

The Histone Demethylase Assay kit (Active Motif) was used to measure formaldehyde production according to the manufacturer's protocol. Synthetic peptides for methylated FOXA2 (258-276 aa) (GenScript, >98% purity) and H3K4me2 (1–21 aa) (Active Motif) were incubated with recombinant LSD1 proteins (Active Motif) at concentrations ranging from 0 to 200 nM in demethylation buffer at 37 °C for 1 h. Subsequently, the reaction mixture was further incubated with the detection buffer for an additional hour at 37 °C, and fluorescence was detected using an excitation wavelength of 410 nm and an emission wavelength of 480 nm.

### Zebrafish embryo metastasis assay

Zebrafish were housed in a fully automated recirculating system in tanks with sizes of either 4 L or 8 L, with a maximum density of ~7 adult fish per liter. System parameters were maintained with a pH of 7.0–7.5 and temperature of 26–27 °C with a 14/10 h light/dark cycle. Water quality was maintained with mechanical and biological filtration, with 5 or more water changes per day, and chemical water quality tests run weekly. Fish were fed 2–3 times per day on a mixed diet. Health was visually monitored daily. Breeding was performed as single pairs or as groups of two or more females per male. Embryos were obtained by natural spawning of AB and Tübingen wild-type lines, and were collected the morning after the fish were set up in breeding containers. All experiments were conducted on 3 day post-fertilization embryos following a protocol approved by the IACUC of University of Massachusetts Boston. High-dose Tricaine (300 mg/L) was used for euthanasia of adult fish and larvae older than 3 days. Hypochlorite (1% bleach solution) following rapid chilling was used for larval fish younger than 3 days. Around 100 GFP-expressing cells were microinjected into the perivitelline space of each embryo. Any fish accidentally injected into the yolk sac were immediately excluded from the study. After the injection, the embryos were washed, transferred to 6-well plates, and imaged with fluorescence microscopy for invasion within an hour after the injection. Embryos exhibiting positive circulation signals were classified as "invaded". The numbers indicated in the figures represent the proportion of "invaded" embryos relative to the total number of injected embryos. The significance of difference was determined by using Fisher's exact test or Chi-squared test depending on sample size.

### Xenograft study

Mouse experiments were conducted in accordance with institutional and U.S. national guidelines and were approved by the IACUC of University of Massachusetts Boston. Male SCID mice were purchased directly from Taconic and housed in shoebox cages with solid floors covered with bedding materials. Upon arrival, the mice were acclimated for 1 week before any experiments were performed, and their health status was monitored daily during this period. Mice were housed at a density of 3-4 mice per cage and fed standard laboratory chow and tap water ad libitum. The housing conditions were maintained at ambient temperatures of 20–24 °C with 40–60% humidity and a 12/12 h light/dark cycle. The cages were changed twice per week by the vivarium staff. During surgery or treatment periods, the mice were observed for signs of pain and distress by well-trained animal staff. To prevent or minimize pain, analgesia was used before and 24 h after surgery. An automated three-stage $CO_2$ delivery system was used for euthanasia in the vivarium.

For the tumor cell injection, PC-3 or NCI-H660 cells were resuspended in serum-free RPMI 1640 medium and mixed with Matrigel (BD Biosciences) in a 1:1 ratio before being subcutaneously implanted ($2 \times 10^6$ cells per injection) on the flanks of castrated male SCID mice (~6 week-old, Taconic). At specified time points, the length (L) and width (W) of the tumors were measured using a caliper, and their volumes were calculated using the formula $L \times W^2/2$. The maximum

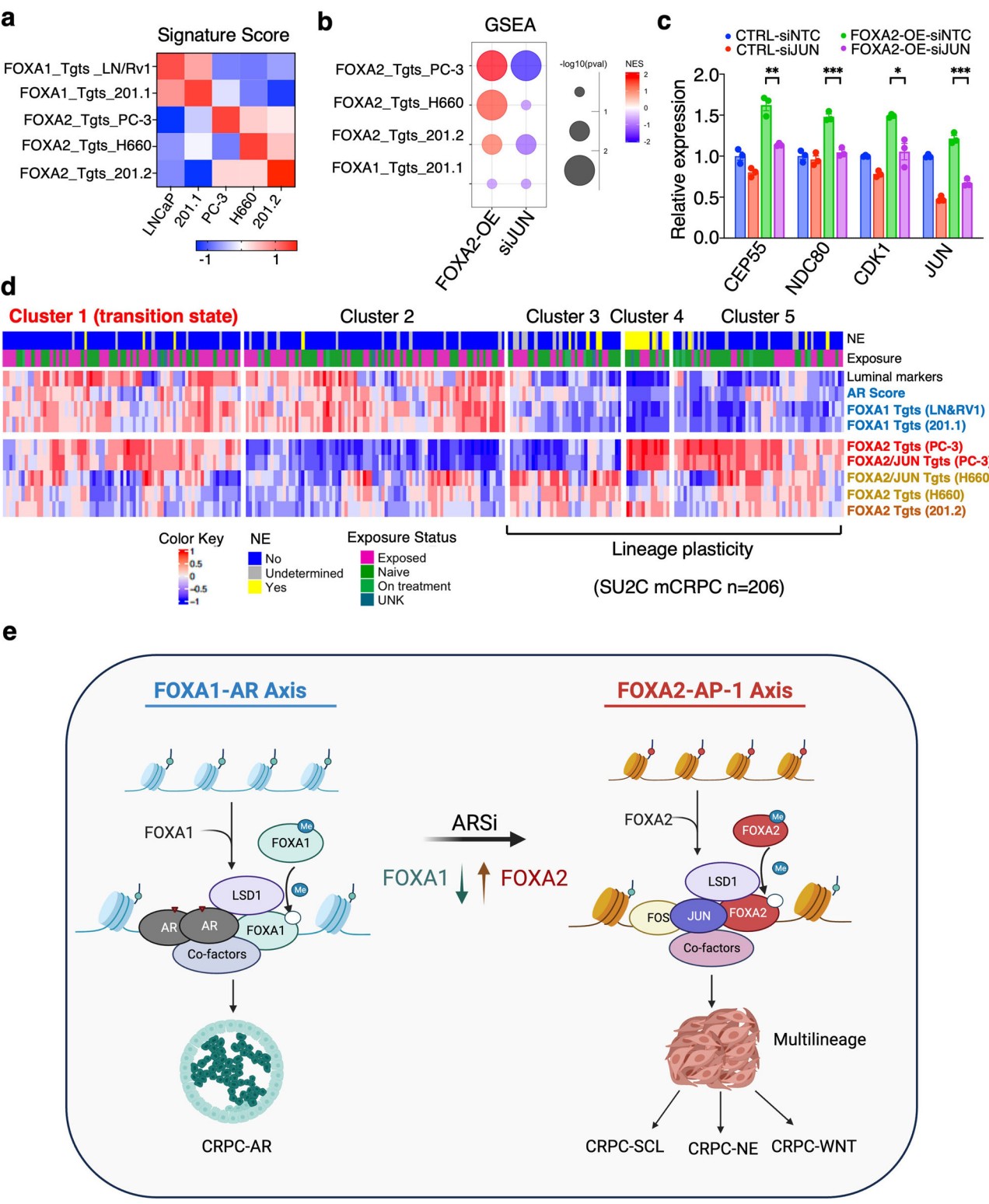

**Fig. 9 | FOXA2 transcription programs are activated in a subset of AR-dependent CRPC tumors. a** Heatmap view for GSVA scores of model-specific FOXA2 targets in different PCa cell lines. **b** GSEA for the enrichment status of indicated FOXA2-target gene sets in LN-FOXA2-OE versus control cells or in LN-FOXA2-OE cells transfected with/out siJUN. Left column: the red color indicates enriched pathways for upregulated genes by FOXA2-OE; Right column: the blue color indicates enriched pathways for genes downregulated by JUN silencing. **c** qRT-PCR for the mRNA expression of indicated FOXA2 targets in the control LNCaP cells and LN-FOXA2-OE cells transfected with/out siJUN (*n* = 3 independent samples; data represented as mean ± SEM; statistical significance determined by unpaired two-sided *t*-test). **d** Heatmap view for GSVA scores of model-specific FOXA2 targets or FOXA2/JUN co-targets in SU2C mCRPC samples (*n* = 206 samples). **e** Graphic model for the functional switch from FOXA1 to FOXA2 in reprogramming JUN and driving lineage plasticity. ns (*P* > 0.05), *(0.01 < *P* < 0.05), **(0.001 < *P* < 0.01), ***(*P* < 0.001), and ****(*P* < 0.0001) were used to indicate the levels of *P*-value. Source data are provided as a Source Data file. **e** Created with BioRender.com released under a Creative Commons Attribution-NonCommercial-NoDerivs 4.0 International license https://creativecommons.org/licenses/by-nc-nd/4.0/deed.en.

allowed tumor size was within 1.5 cm in any dimension according to the approved IACUC protocol. For the experiment with FOXA2 silencing, we defined the day of cell injection as day 0. For the experiments involving drug treatment, we allowed average tumor size to reach 100 mm³ and then considered the first day of treatment as day 0.

### Statistics and reproducibility

All ChIP-qPCR and qRT-PCR data were presented in the bar graph format from samples collected from 3 independent tissue cultures. Statistical analysis was generally performed using unpaired two-tailed Student's $t$-test (parametric test) by comparing treatment versus vehicle control or otherwise as indicated. For animal studies, a two-way ANOVA test was performed to determine the statistical difference in tumor growth at the final time point. We use ns ($P > 0.05$), *($0.01 < P < 0.05$), **($0.001 < P < 0.01$), ***($P < 0.001$), and ****($P < 0.0001$) to indicate the levels of $P$-value. The results for immunoblotting are representative of at least three biologically independent experiments showing similar results. All statistical analyses and visualization were performed by using GraphPad (Prizm 8) or R (version 3.4.0) unless otherwise specified.

### Reporting summary

Further information on research design is available in the Nature Portfolio Reporting Summary linked to this article.

## Data availability

The ChIP-seq, ATAC-seq, and RNA-seq data generated in this study have been deposited in the Gene Expression Omnibus (GEO) database under accession code GSE232555. The protein mass-spectrometry data have been deposited in ProteomeXchange database (PXD) under accession code PXD052273. The ChIP-seq publicly available data used in this study are available in the GEO database under accession code GSE52201[29], GSE114268[26], GSE72467[47], GSE137209[60], and GSE77448[61]. The RNA-seq publicly available data are under accession code GSE8702[48]. The remaining data are available within the Article, Supplementary Information or Source Data file. Source data are provided with this paper.

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

## Acknowledgements

This work is supported by grants from NIH (R01 CA211350 to C.C. and U54 CA156734 to C.C. and J.A.M.), DOD (W81XWH-19-1-0361 and W81XWH-21-1-0267 to C.C., and W81XWH-19-1-0777 to S.G.), CIHR (142246 and 159567 to H.H.H.), Cancer Council SA (2012127 to L.A.S.), Cancer Council NSW (2020404 to L.A.S.), Movember & National Breast Cancer Foundation Collaboration Initiative grant (MNBCF-17-012 to G.P.R. and L.A.S.), Department of Health and Human Services acting through the Victorian Cancer Agency (fellowships to M.G.L. MCRF18017, R.A.T. MCRF15023), the EJ Whitten Foundation, Movember (GAP Xenografting Consortium), the Peter and Lyndy White Foundation, TissuPath Pathology, and the Rotary Club of Manningham. L.A.S. is supported by a Principal Cancer Research Fellowship (PRF2919) awarded by Cancer Council's Beat Cancer project on behalf of its donors, the State Government through the Department of Health and the Australian Government through the Medical Research Future Fund. C.C. and D.H. were supported by Proposal Development Award from University of Massachusetts Boston. Z.W., M. Liu., and M. Li were supported by CSM (College of Science and Mathematics) Dean's Doctoral Research Fellowship from the University of Massachusetts Boston. M. Liu was supported by a graduate fellowship from the Integrative Biosciences Program at the University of Massachusetts Boston. We thank Camille OrtizCerezo, Yuhan Liu, Nolan Pattern, Krishna Patel, and Michaela Mulhearn for their assistance in various molecular assays and sequencing studies, Rachel Muriph and Dr. Evens for their assistance in the mass-spectrometry study, and other members of the Cai, Selth, Lawrence, and He laboratories for constructive input. We also thank Dr. Leigh Ellis for kindly providing SKO/DKO cell lines. We acknowledge the patient representatives, clinical coordinators, scientists, and clinicians, who contribute to the Melbourne Urological Research Alliance (MURAL) and its collection of patient-derived models.

## Author contributions

C.C., Z.W., H.H.H., and L.A.S. designed the study. Z.W., S.L.T., S.Z., M. Liu, M. Li, M. Labaf, K.V., K.R.S., D.H., G.P.R., and R.A.T. performed experiments and analyzed the results. Z.W., S.L.T., S.P., and J.A.M. performed deep sequencing analyses. C.C., Z.W., S.G., M.G.L., and L.A.S. wrote and revised the manuscript. All authors discussed the results and commented on the manuscript.

## Competing interests

G.P.R, R.A.T. and M.G.L. have research collaborations with AstraZeneca, Pfizer. The remaining authors declare no competing interests.

## Ethics

Animal experiments were conducted in accordance with institutional and USA national guidelines and were approved by the Institutional Animal Care and Use Committee (IACUC) of University of Massachusetts Boston. Animal husbandry is provided by the Animal Resources Core Facility and Vivarium. The facility is under the direction of licensed veterinarians and complies with federal and state regulations and guidelines for humane care and use of vertebrate animals in research. Animal group sizes were estimated based on the power analysis using preliminary data and were approved by IACUC.
