## [Peer Review File · Nature Communications]

FOXA2 rewires AP-1 for transcriptional reprogramming and lineage plasticity in prostate cancerReviewers' Comments:

Reviewer #1:

Remarks to the Author:

FOXA (Forkhead Box Protein A) family proteins play essential roles in pioneering transcription activities. Specifically, FOXA1-mediated AR signaling and gene regulatory networks are well-established in both primary and castration-resistant prostate cancer (CRPC). While recent studies have highlighted the oncogenic functions of FOXA2 in neuroendocrine PCa, the detailed molecular and genomic roles of FOXA2 during PCa progression remain less explored. In this manuscript, Wang et al. characterized the chromatin landscapes of FOXA2 using various PCa cell lines and patient-derived xenograft models. Their findings suggest that FOXA2 bindings are lineage-specific and correlate with distinct PCa molecular subtypes. They also showed that the genomic activities of FOXA2 are predominantly mediated by LSD1. This is evidenced by the disrupted chromatin associations of FOXA2 when influenced by LSD1 inhibitors and LSD1-directed K265 methylation of FOXA2 in gene regulation. Moreover, the team identified JUN as a key collaborating factor with FOXA2, helping orchestrate PCa subtype-specific binding profiles and activate lineage-specific super enhancers. The study concludes by noting that FOXA2 overexpression enhances the multilineage plasticity of primary PCa cells and coincides with a chromatin redistribution of JUN. Using FOXA1/FOXA2 gene signatures, they stratified metastatic CRPC patients into five molecular clusters. Overall, this study presents some interesting points; however, there are areas that could be further clarified.

1. Different antibodies used in ChIP-seq can lead to varied binding. Have the authors considered this when comparing FOXA1 and FOXA2 binding, especially when claiming distinct binding sites for both in Fig. 1?
2. The authors have compared FOXA1 and FOXA2 binding across different cell types. Since NCI-H660 cells express both FOXA1 and FOXA2, could the authors juxtapose their binding profiles to emphasize the unique genomic activities of FOXA2 within the same genetic background?
3. The conclusion in Fig. 6, suggesting FOXA2 expression in AR-dependent PCa cells catalyzes a multilineage progression, needs more robust computational and experimental biology data for support. Are the three non-AR CRPC subtype gene signatures (Tang et al. Science 2022) enriched in FOXA2 overexpressed LNCaP cells? Does FOXA2 overexpression induce expression of subtype-specific gene markers (Tang et al. Science 2022)? It's also worth noting that Jun binding at the non-AR subtype ATAC sites, especially the SCL sites, appears weak.
4. To solidify the notion that FOXA2-directed gene transcription is influenced by LSD1, an analysis comparing transcriptomic shifts and JUN bindings in LNCaP cells overexpressing the FOXA2 K265 mutant with those from the FOXA2 wild-type would be beneficial.

Reviewer #2:

Remarks to the Author:

In this study Wang et al., employed prostate cell lines and patient-derived xenograft models and revealed that lysine demethylase LSD1 determined FOXA2 chromatin binding and transcription program and JUN as a major cooperative transcription factor of FOXA2 in AR-null PCa models. The study is very well written and presented but some conclusions were overstated and lacked experimental verification.

1. In clinical, most of CRPC patients develop from prostate cancer treated with androgen deprivation therapies. As the authors have used cell lines representing different lineages, can they comment on the effects of AR signaling inhibitors for different lineages on the expression and the chromatin binding of LSD1/FOXA2/AP-1 family?
2. The authors showed two potent LSD1 inhibitors, ORY1001 and C12, could disrupted the association of FOXA2 with chromatin in PC-3 and NCI-H660 cells, however, these inhibitors may also impact other proteins' function. Gene-silencing or overexpression of LSD1 then examine methylation and chromatin binding of FOXA2 will be critical. Also, overexpression of LSD1 in the condition of FOXA2-K265R is

necessary.

3. As shown in Fig. 2g, chromatin binding of the mutant FOXA2 was increased with ORY-1001 treatment, how does the author explain for the increase? Is this because of the non-specificity of the ORY-1001?

4. In Fig.2 and Extended Data Fig.3, the authors only showed the silence of FOXA2 decreased the DNPC cell line growth, migration, and invasion. LSD1 inhibition repressed the proliferation and migration of PC-3 elicited by the overexpression of FOXA2. How about the NEPC cell line, NCI-H660?

5. The authors claimed that LSD1 could act as a binding partner of FOXA1 and FOXA2. What are the similarities and differences of the LSD1 binding pattern in ARPC, DNPC and NEPC cell lines?

6. The authors claimed that FOXA2 in cooperation with JUN, can mediate CRPC lineage plasticity and they validated that FOXA2 overexpression in LNCaP cells initiates a multilineage progression. However, they didn't show the necessity of existence of JUN in mediating lineage plasticity. Overexpression of FOXA2 and silencing JUN simultaneously in ARPC cell lines (in vitro and in vivo) then examine the expression of the lineage-specific markers is vital.

7. As the authors claimed, JUN mRNA levels were similar in AR-high and AR-low tumors. Does it mean that the ARPC cell line such as LNCaP can also respond to JUN inhibitors? Are PC-3 and NCI-H660 more sensitive than LNCaP for JUN inhibitors?

8. The clinical implication of AP-1 family targeting seems to be challenging. As AP-1 family widely expresses and functions in various organs and tissues. How do the authors evaluate it?

9. The authors stated that the epigenetic axis, LSD1-FOXA2 could promote PCa lineage plasticity. However, they only verified the function of FOXA2 in mediating CRPC lineage plasticity. LSD1 regulating lineage plasticity is not fully validated. Gene-silencing of LSD1 then examine methylation of FOXA2 and lineage changes in DNPC and NEPC cell lines will be important.

10. The authors reported that FOXA2 overexpression increased the proliferation and migration of LNCaP cells under castrated conditions. How does the AR expression change with FOXA2 silencing in NCI-H660? Will it be sensitive for androgen deprivation treatment?

11. The authors revealed that LSD1 determined FOXA2 chromatin binding and transcription program and JUN as a major cooperative transcription factor of FOXA2 only in prostate cell lines. Whether these conclusions apply for human samples and mouse models, such as p53^{-/-};Rb1^{-/-} mouse tissues from lung and prostate? The in validation using in vivo models is needed.

Reviewer #3:

Remarks to the Author:

Reviewer #4:

Remarks to the Author:

The Ms by Cai and co-Authors entitled "FOXA2-LSD1 epigenetic axis rewires AP-1 for transcriptional reprogramming lineage plasticity in prostate cancer" reports preclinical results that are very interesting to deepen the knowledge on the prostate cancer plasticity that induce cancer cells to progress to CRPC and thereafter into further degeneration, such as the neuroendocrine PCa. Introduction and Discussion are well written and readable, while the Methods, Results, Figures and Legends are very difficult to comprehend, with criticisms in Methods, Results and Figures, regarding in particular experiments done in animals (both in the main Ms and in the Extended data).

Methods are not well described and they do not give enough information, in particular concerning the in vivo experiments. No information on the LSD1 inhibitors (also for the in vitro experiments), no information about cell treatment protocol (10 μ M for 24hours based on...?), no information on animal

numbers.

Results: Results are too many and too compact into the Figures. Figures have too many inserts (more than 10), often with discrepancies in the citation between Figures, Legend and Results.

Major criticisms:

- Fig. 2, inserts k and m and Methods (lines 756-759): Experiments with zebrafish embryos

What are the TUE zebrafish lines? Why have they been used, besides AB? How many embryos were injected? What do the numbers in the Figure inserts mean?

Then, the major criticism is related to the Figure reported: Although PC-3 are of metastatic origin, the experimental approach using 100 cells injected in the perivitelline space, an avascular space, taking a picture 1 hour after injection is difficult to comprehend. Authors need to explain the discrepancy with other published papers, in which higher number of PC-3 cells and much more longer time-course points are used (i.e. doi: 10.1038/s41388-019-1103-0; 10.1038/s41598-019-44452-4) or, more in general, with the methodological approach generally used for these experiments (reviewed in doi: 10.3390/cells9091978; 10.1016/j.bbdis.2013.01.016). Images are of poor resolution, the shFOXA2 embryo has cells injected in the yolk, not in the perivitelline sac, while pictures of the embryos in the veh and ORY image present a very high background fluorescence, that does not allow to see the cell mass and that could mask the presence of metastasis. Furthermore, the sagittal alignment is not correct, i.e. shFOXA2 embryo displayed both eyes. Finally, a minor point concerns how the zebrafish embryos are usually depicted: indeed, present figures need to be rotated by 180°, as conventionally, zebrafish embryos are shown with the head on the left and the yolk facedown

- Fig.2j-l, Fig. 4i-j: Experiments with SCID mice and Methods (lines 762-767)

Authors need to clarify the timing of tumor volume measurement, as it seems that after injection of 2×10^6 wt PC-3 cells (very low number compared to published literature), in 6 days (showing an unsuspected very high proliferation rate compared to other reports) the tumor volume reaches about 100 mm³ volume (Fig.2j), while in Fig.2l and Fig. 4 I - j, at day 0, the volume is already about 100mm³

- Many western blot images are of poor quality. No one has been quantified

Minor points:

Why Authors decided to xenograft PC-3 cells in zebrafish embryos to study the metastasis formation and in nude mice to study the tumor volume?

Fig.7: I do understand the significance of CRPC-SCL and CRPC-NE, while what the acronym CRPC-WNT means?

Summary of the Revision: We are thankful for the reviewers' enthusiasm and acknowledgment of the significance of our work. In response to their comments, we have reviewed each point and provided detailed, point-by-point responses in this letter. Over the past six months, we have generated a substantial amount of additional data, which is now included in the revised figures. In particular, we have significantly strengthened the major findings by conducting whole-genomic analyses and further validations in the *Pten*^{-/-}/*Rb*^{-/-} murine DKO model, which expresses high levels of FOXA2, LSD1, and JUN (new **Supplementary Figure 8a-i**), and studies using methylation-deficient K265R mutant of FOXA2 (new **Supplementary Figure 13b-g**). We believe that these new findings not only support but also significantly bolster the major conclusions of our study.

Reviewers' Comments:

Reviewer #1: Comments:

FOXA (Forkhead Box Protein A) family proteins play essential roles in pioneering transcription activities. Specifically, FOXA1-mediated AR signaling and gene regulatory networks are well-established in both primary and castration-resistant prostate cancer (CRPC). While recent studies have highlighted the oncogenic functions of FOXA2 in neuroendocrine PCa, the detailed molecular and genomic roles of FOXA2 during PCa progression remain less explored. In this manuscript, Wang et al. characterized the chromatin landscapes of FOXA2 using various PCa cell lines and patient-derived xenograft models. Their findings suggest that FOXA2 bindings are lineage-specific and correlate with distinct PCa molecular subtypes. They also showed that the genomic activities of FOXA2 are predominantly mediated by LSD1. This is evidenced by the disrupted chromatin associations of FOXA2 when influenced by LSD1 inhibitors and LSD1-directed K265 methylation of FOXA2 in gene regulation. Moreover, the team identified JUN as a key collaborating factor with FOXA2, helping orchestrate PCa subtype-specific binding profiles and activate lineage-specific super enhancers. The study concludes by noting that FOXA2 overexpression enhances the multilineage plasticity of primary PCa cells and coincides with a chromatin redistribution of JUN. Using FOXA1/FOXA2 gene signatures, they stratified metastatic CRPC patients into five molecular clusters. Overall, this study presents some interesting points; however, there are areas that could be further clarified.

Response: We very much appreciate the reviewer's positive comments on our study and trust that our responses below, in combination with the new data, clarify the reviewer's queries in a satisfactory manner.

1. Different antibodies used in ChIP-seq can lead to varied binding. Have the authors considered this when comparing FOXA1 and FOXA2 binding, especially when claiming distinct binding sites for both in Fig. 1?

Response: We appreciate the feedback from the reviewer. In response to the raised concern, we conducted additional ChIP-seq experiments for FOXA2 using various antibodies and replicates in both PC-3 cell line and 201.2 PDX models. The results, as shown in the new **Supplementary Figure 1a-e**, demonstrate a high degree of consistency in chromatin binding across different antibodies. Notably, in PC-3 cells, FOXA2 binding consistently showed enrichment at sites associated with the SCL (stem cell-like) signature, whereas in 201.2 tumors, there was a consistent enrichment at WNT signature sites. Although our peak calling methodology might yield varying numbers in the Venn diagrams, the identification of distinct chromatin binding patterns is clearly demonstrated in the heatmap visualizations. Despite the potential small variation in the number of distinct peaks with the use of different antibodies, we believe the overall conclusion remains robust and valid.

2. The authors have compared FOXA1 and FOXA2 binding across different cell types. Since NCI-H660 cells express both FOXA1 and FOXA2, could the authors juxtapose their binding profiles to emphasize the unique genomic activities of FOXA2 within the same genetic background?

Response: It would be intriguing to explore how FOXA1 chromatin binding is influenced by FOXA2 expression in NCI-H660 cells. Unfortunately, we have encountered a limitation with commercially available ChIP-grade FOXA1 antibodies, which cross-react with FOXA2, thus detecting both FOXA2 expression and FOXA2 chromatin binding. This issue is demonstrated in **Figure R1**, provided here for the reviewer's reference. While this cross-reactivity is not a significant concern when performing FOXA1 ChIP-seq in FOXA2-null cells, such as LNCaP, CWR22-RV1, and PDX201.1 models, it leads to incorrect interpretations in FOXA2-high cells. This effect may be even more amplified in ChIP-seq analyses. For example, we performed FOXA1 ChIP-seq in PC-3 cells, which should have very low/undetectable FOXA1 expression, and observed a large number of binding peaks which are almost identical to our FOXA2 ChIP-seq results (**Figure R2**, for reviewer only). This suggests that FOXA1 ChIP-seq in cells with high FOXA2 expression is not reliable. Therefore, although investigating FOXA1 binding in NCI-H660 cells is compelling, these technical difficulties prevent a straightforward analysis. However, we have repeatedly validated the specificity of our FOXA2 antibodies, confirming it does not cross-react with FOXA1 (see Figure R1). Thus, all FOXA2 bindings observed in our models, including the NCI-H660 cell line, are specific to FOXA2.

3. The conclusion in Fig. 6, suggesting FOX A2 expression in AR-dependent PCa cells catalyzes a multilineage progression, needs more robust computational and experimental biology data for support. Are the three non-AR CRPC subtype gene signatures (Tang et al. Science 2022) enriched in FOX A2 overexpressed LNCaP cells? Does FOX A2 overexpression induce expression of subtype-specific gene markers (Tang et al. Science 2022)? It's also worth noting that Jun binding at the non-AR subtype ATAC sites, especially the SCL sites, appears weak.

Response: We thank the reviewer for this excellent suggestion. In light of this, we have conducted additional analyses. As shown in the newly added **Figure 8g**, there is a significant increase in the expression of NE and WNT transcriptional signatures. Interestingly, despite the clear enrichment of FOX A2 binding at SCL signature sites, the SCL transcription signature does not show any increase. This observation aligns with the altered JUN chromatin binding, which appears more enriched at NE and WNT sites rather than SCL sites. These findings imply that the multilineage reprogramming initiated by FOX A2 may be further influenced by additional mechanisms, particularly in the regulation of AP-1 reprogramming. We have updated both the results and discussion sections of our manuscript to incorporate these new insights.

4. To solidify the notion that FOX A2-directed gene transcription is influenced by LSD1, an analysis comparing transcriptomic shifts and JUN bindings in LNCaP cells overexpressing the FOX A2 K265 mutant with those from the FOX A2 wild-type would be beneficial.

Response: We appreciate the constructive suggestions from the reviewer. In response, we have generated an additional LNCaP cell line overexpressing **3xFLAG-FOX A2-K265R** mutant, as shown in the new

Supplementary Figure 13b. We then conducted a ChIP-seq analysis of JUN in these cells to compare its chromatin binding with that in wild-type cells. As shown in **Supplementary Figures 13c** and **d**, there was a notable expansion in JUN binding and some redistribution of this binding. More significantly, we observed an enhanced enrichment of JUN binding at all three non-AR signature sites, particularly at NE and WNT signature sites (**Supplementary Figure 13e**). Additionally, we confirmed increased expression of some FOXA2/JUN targets by qRT-PCR (**Supplementary Figure 13f**). We next performed RNA-seq in these cells and analyzed lineage-specific transcriptional signatures. The findings indicate that NE and WNT signatures are significantly elevated by the K265R mutant (SCL scores also increased, but not statistically significant) (**Supplementary Figure 13g**).

Surprisingly, the FOXA2-K265R mutant, unlike the WT FOXA2, did not suppress endogenous FOXA1 expression (**Supplementary Figure 13b**). This unexpected finding suggests a more complex interplay between LSD1, FOXA1 and FOXA2 that requires further investigation but is beyond the scope of this current study. Accordingly, we have revised our manuscript to address this issue and have slightly altered our title and abstract, removing the statement that LSD1 promotes lineage plasticity. Nonetheless, these results do not alter our major conclusion regarding LSD1-enhanced FOXA2 chromatin binding and transcription activity.

Reviewer #2: Comments:

In this study Wang et al., employed prostate cell lines and patient-derived xenograft models and revealed that lysine demethylase LSD1 determined FOXA2 chromatin binding and transcription program and JUN as a major cooperative transcription factor of FOXA2 in AR-null PCa models. The study is very well written and presented but some conclusions were overstated and lacked experimental verification.

Response: We appreciate very much these positive comments from this reviewer and have modified our text and provided additional data to verify some key conclusions.

1. In clinical, most of CRPC patients develop from prostate cancer treated with androgen deprivation therapies. As the authors have used cell lines representing different lineages, can they comment on the effects of AR signaling inhibitors for different lineages on the expression and the chromatin binding of LSD1/FOXA2/AP-1 family?

Response: Thank you for bringing this matter to our attention. First, we want to indicate that we have cited in the manuscript the recently published study from Han et al (PMID: 36332622), which showed that androgen deprivation treatment (ADT) in the *Pten^{-/-}IRb^{-/-}ITrp53^{-/-}* murine PCa model significantly promoted FOXA2 expression and its chromatin binding. To expand on this, we also analyzed public databases, focusing on how androgen-dependent human PCa cells transition to an androgen-insensitive state. Our findings, now included in new **Supplementary Figure 12a** and **b**, reveal that LNCaP cells under prolonged ADT exhibit a multilineage progression and a gradual increase in FOXA2 expression, compared to their androgen-dependent parental cells. Based on these evidence, we hypothesize that ARSi treatments such as enzalutamide might more potently elicit this multilineage transition, although we did not find any high-quality published H3K27ac data in relevant models to confirm it. Our current understanding suggests that ARSi likely triggers FOXA2 expression to suppress FOXA1 and the luminal lineage and then reprogram AP-1 to lead the multilineage transition. This process requires LSD1-mediated demethylation of FOXA2 to stabilize its chromatin binding. Subsequent expression/activation of lineage-specific factors, including NE drivers (such as ASCL1 and NEUROD1) or DNPC drivers (such as FGF/WNT signaling), is likely to play a crucial role in further determining the final trajectory of lineage reprogramming.

2. The authors showed two potent LSD1 inhibitors, ORY1001 and C12, could disrupted the association of FOXA2 with chromatin in PC-3 and NCI-H660 cells, however, these inhibitors may also impact other proteins' function.

Gene-silencing or overexpression of LSD1 then examine methylation and chromatin binding of FOXA2 will be critical. Also, overexpression of LSD1 in the condition of FOXA2-K265R is necessary.

Response: In response to the reviewers' suggestions, we conducted experiments using specific siRNAs to knock down LSD1. The results, as shown in new **Figure 3j**, indicate that silencing LSD1 leads to an increase in the methylation of FOXA2, consistent with observations made using LSD1 inhibitors. Further investigations were carried out to assess the impact of LSD1 silencing on FOXA2 chromatin binding. The findings, demonstrated in new **Figures 3f-h**, reveal a significant reduction in FOXA2 chromatin binding across various target sites upon LSD1 silencing. In addition, we explored the effects of LSD1 silencing in the context of the K265R mutation of FOXA2. Our data show that the suppressive effect of LSD1 silencing on FOXA2 chromatin binding is diminished in cells expressing the K265R mutant (new **Figure 3n**), mirroring the pattern observed with LSD1 inhibitors. Globally using RNA-seq, we further demonstrated that siLSD1 decreases the FOXA2 transcription program in PC-3 cells, similar to LSD1 inhibition (new **Figure 4a** and **4c**). Collectively, these findings reinforce our conclusion that LSD1 plays a crucial role in stabilizing FOXA2 chromatin binding and enhancing its transcriptional activity.

3. As shown in Fig. 2g, chromatin binding of the mutant FOXA2 was increased with ORY-1001 treatment, how does the author explain for the increase? Is this because of the non-specificity of the ORY-1001?

Response: Thank you for raising this intriguing question. Our observations have been consistent across experiments using both siLSD1 and the LSD1 inhibitor (see revised **Figure 3m** and **n**), suggesting that the effect is not a result of non-specific actions of ORY-1001. We propose that this phenomenon may be explained by the interplay between LSD1 inhibition/silencing and alterations in its histone substrate H3K4me. Specifically, as expected and now shown in new **Figure 3h**, LSD1 silencing is associated with an increase in H3K4me2 levels. This increase in H3K4me2, which has been previously shown to enhance FOXA1 chromatin binding (PMID 18358809), may also positively influence FOXA2 chromatin binding. In the scenario of the overexpression of FOXA2-K265R mutant, although LSD1 inhibition or silencing does not decrease the chromatin binding of mutant FOXA2, it still leads to elevated H3K4me2 levels. This alteration could potentially create a feedback mechanism that results in the overall enhancement of mutant FOXA2 chromatin binding.

4. In Fig.2 and Extended Data Fig.3, the authors only showed the silence of FOXA2 decreased the DNPC cell line growth, migration, and invasion. LSD1 inhibition repressed the proliferation and migration of PC-3 elicited by the overexpression of FOXA2. How about the NEPC cell line, NCI-H660?

Response: In response to the comment, we have conducted further experiments to evaluate the effects of LSD1 inhibition/silencing and FOXA2 silencing on NCI-H660 cells. The results, which are now included in the new **Supplementary Figures 4f** and **4g**, demonstrate that both LSD1 inhibition/silencing and FOXA2 silencing lead to a comparable reduction in cell proliferation in NCI-H660 cells. Regarding the evaluation of migration ability, it is important to note that NCI-H660 cells exhibit very low baseline migration capabilities due to their suspension cell culture characteristics. Therefore, we did not test the migration in vitro in this model.

5. The authors claimed that LSD1 could act as a binding partner of FOXA1 and FOXA2. What are the similarities and differences of the LSD1 binding pattern in ARPC, DNPC and NEPC cell lines?

Response: In our manuscript, we presented the binding patterns of LSD1 in LNCaP cells (ARPC) and PC-3 cells (DNPC), demonstrating their correlation with lineage-specific signature sites (originally in Figure 2a, now revised to **Figure 3b**). Our previous studies have also demonstrated strong overlap of FOXA1 and LSD1

global chromatin binding in LNCaP cells (PMID25482560; PMID32868907). The FOXA2 and LSD1 binding were highly associated in PC-3 cells, demonstrated in this study (see **Figure 3a**). However, we have not been able to obtain high-quality ChIP-seq for LSD1 in other models due to technical difficulties.

It is important to note that these LSD1 ChIP-seq data were obtained from prior experiments. We have put significant effort to conducting LSD1 ChIP-seq in NCI-H660 cells and 201.1/2 PDX models. However, we encountered a major challenge with the commercially available LSD1 antibodies. Despite testing four different antibodies (Abcam ab17721, ab129195, CST 2139S, and Millipore CS207350), none yielded high-quality ChIP-seq results. For the reviewer’s information, the ChIP-seq performed in NCI-H660 cells (using a mixture of antibodies from Cell Signaling 2139s and Millipore CS207350) generated very few binding peaks, especially when compared to LNCaP cells, where a specifically designed antibody was used (this antibody is unfortunately no longer available for this study) (see **Figure R3a**, for reviewer reference only). Interestingly, despite the low quality of these results, we still observed a correlation between LSD1 binding peaks and FOXA2 and ATAC-seq signals in the NCI-H660 models (**Figure R3b**, for reviewer reference only). Additionally, these LSD1 binding peaks appear to be enriched in CRPC-NE signature sites (**Figure R3c**, for reviewer reference only). While these findings do support our hypothesis that LSD1 chromatin binding correlates with FOXA2 and is enriched in NE, we have decided not to include this data in the main manuscript due to the low quality of the dataset.

6. The authors claimed that FOXA2 in cooperation with JUN, can mediate CRPC lineage plasticity and they validated that FOXA2 overexpression in LNCaP cells initiates a multilineage progression. However, they didn’t show the necessity of existence of JUN in mediating lineage plasticity. Overexpression of FOXA2 and silencing JUN simultaneously in ARPC cell lines (in vitro and in vivo) then examine the expression of the lineage-specific markers is vital.

Response: In line with the reviewer’s suggestion, we conducted RNA-seq on LN-FOXA2-OE cells following JUN silencing. The results support our conclusions, showing that JUN silencing reduces the FOXA2

transcription programs across multiple lineages (new **Figures 9b** and **c**). Notably, the most pronounced effects were observed in PC-3 targets (representing the SCL model) and 201.2 targets (representing the WNT model).

7. As the authors claimed, JUN mRNA levels were similar in AR-high and AR-low tumors. Does it mean that the ARPC cell line such as LNCaP can also respond to JUN inhibitors? Are PC-3 and NCI-H660 more sensitive than LNCaP for JUN inhibitors?

Response: To address the question, we assessed the response of cell growth to the AP-1 inhibitor in LNCaP and PC-3 cells. The results, now shown in new **Figure 6i**, demonstrate that PC-3 cells are more sensitive to AP-1 inhibition compared to LNCaP cells. Additionally, we examined the effect of AP-1 inhibitors on a pair of murine PCa models, SKO and DKO, which are respectively FOXA2 negative and positive, while both expressing JUN (the rationale for this comparison is detailed in the response below for Q11). Our findings, presented in new **Supplementary Figure 8i**, reveal that the SKO cells exhibit resistance to AP-1 inhibitor treatment, whereas the DKO cells are sensitive with an IC50 of less than 1 μ M. These observations suggest that FOXA2-positive PCa tumors have increased sensitivity to AP-1 inhibition, while ARPC cells tend to be generally insensitive to AP-1 inhibition.

8. The clinical implication of AP-1 family targeting seems to be challenging. As AP-1 family widely expresses and functions in various organs and tissues. How do the authors evaluate it?

Response: We absolutely agree with the reviewer that targeting AP-1 could lead to other effects beyond the intended therapeutic outcomes. However, recent studies have indicated that tumor cells may develop a dependency on certain oncogenic transcription pathways (such as pathways involving the regulation of Super-Enhancers). This suggests the possibility of a therapeutic window where specific doses of inhibitors could be effective without causing harm. Notable examples include BRD4 and EZH2, which, despite being ubiquitously expressed in various organs and tissues, can be targeted by inhibitors to yield promising effects in tumor treatment. Notably, an EZH2 inhibitor has already received FDA approval. Given the role of AP-1 in regulating Super-Enhancers shown in this manuscript, we hypothesize that AP-1 inhibitors could similarly have a dosage window that allows effective treatment of AR-independent PCa while minimally impacting normal bodily functions. This concept aligns with the idea that selective inhibition of oncogenic pathways in tumors, even those involving broadly expressed factors, can be therapeutically beneficial.

9. The authors stated that the epigenetic axis, LSD1-FOXA2 could promote PCa lineage plasticity. However, they only verified the function of FOXA2 in mediating CRPC lineage plasticity. LSD1 regulating lineage plasticity is not fully validated. Gene-silencing of LSD1 then examine methylation of FOXA2 and lineage changes in DNPC and NEPC cell lines will be important.

Response: We are grateful for this constructive suggestion. As mentioned in our response to question #2, we have shown that silencing LSD1 leads to an increase in FOXA1 methylation (see new **Figure 3j**). To further explore the impact on lineage reprogramming, we investigated the effects of LSD1 inhibition on the transcriptional signatures of different lineages. The results, presented in new **Supplementary Figures 3d** and **3e**, reveal a notable decrease in the CRPC-SCL signature coupled with an increase in the CRPC-AR signature in PC-3 cells. Additionally, we observed a significant reduction in the CRPC-NE signature in NCI-H660 cells. These findings suggest that LSD1 indeed plays a role in maintaining these specific lineages of PCa cells. However, given the intricate role of LSD1 in modulating FOXA1 and FOXA2 in the LN-FOXA2-OE model (as detailed in our response to reviewer 1's question 4), the precise function of LSD1 in the FOXA2-

mediated multilineage transition in PCa remains ambiguous. Therefore, we have modified our text to exclude the assertion that LSD1 directly regulates lineage plasticity.

10. The authors reported that FOXA2 overexpression increased the proliferation and migration of LNCaP cells under castrated conditions. How does the AR expression change with FOXA2 silencing in NCI-H660? Will it be sensitive for androgen deprivation treatment?

Response: We investigated the expression of AR and AR signaling signature genes in AR-null NCI-H660 and PC-3 cells. The results, shown in new **Supplementary Figures 3c** and **3d**, indicate that silencing FOXA2 does not significantly affect the expression of AR gene or its target genes in NCI-H660 cells. In PC-3 cells, while silencing FOXA2 did lead to an increase in AR expression, it did not result in an enhancement of AR signaling. Therefore, it is highly unlikely that these cells can become sensitive to ADTs or ARSi. However, it is conceivable that in a clinical setting, targeting FOXA2 early might resensitize tumors to AR signaling inhibitors (ARSi), especially in the presence of AR-dependent tumor clones.

11. The authors revealed that LSD1 determined FOXA2 chromatin binding and transcription program and JUN as a major cooperative transcription factor of FOXA2 only in prostate cell lines. Whether these conclusions apply for human samples and mouse models, such as p53^{-/-};Rb1^{-/-} mouse tissues from lung and prostate? The in validation using in vivo models is needed.

Response: We greatly appreciate the reviewer's suggestion and, in response, have chosen to utilize a pair of murine PCa models, SKO and DKO, for further validation. Prostate tumor cell were derived from PCa tumors in *Pten*^{-/-} (SKO) and *Pten*^{-/-}/*Rb*^{-/-} (DKO) mice, respectively (Ku et al., PMID 28059767). While both the SKO and DKO models express Jun, only DKO cells are positive for Foxa2 (see new **Supplementary Figure 8a**). We also attempted to test the *Pten*^{-/-}/*Rb*^{-/-}/*TP53*^{-/-} TKO cells but they cannot maintain stable Foxa2 protein expression under culture conditions.

Nonetheless, subsequent RNA-seq analysis was performed on DKO cells with silenced FOXA2 (new **Supplementary Figure 8b**). The results, demonstrated in new **Supplementary Figure 8c**, reveal that Foxa2-upregulated genes are significantly enriched in AP-1 pathways. Further examination of lineage signatures indicates a notable enrichment of Foxa2-activated genes in the CRPC-SCL transcriptional signature (new **Supplementary Figure 8d**), and predicts a correlation between Foxa2 transcriptome and Fosl1 chromatin binding (new **Supplementary Figure 8e**). This implies that DKO cells may exhibit stem cell-like characteristics.

To further validate Lsd1 activity on Foxa2 chromatin binding, we performed ChIP-seq analysis of Foxa2 in DKO cells. We identified 5,715 high-confidence peaks significantly associated with Foxa2-activated genes (new **Supplementary Figure 8f**). Foxa2 binding sites were also highly enriched for FOXA and AP-1 motifs (new **Supplementary Figure 8g**). Importantly, the global binding of Foxa2 was dramatically suppressed by Lsd1 inhibition, consistent with our observations in human PCa models (new **Supplementary Figure 8h**). We then validated the LSD1-mediated Foxa2 chromatin stabilization and FOXA2-mediated Jun binding at two enhancer sites located at the *Mmp3* and *Mmp13* genes (new **Supplementary Figures 8i** and **j**). Moreover, our results demonstrate that DKO cells are hypersensitive to LSD1 or AP-1 inhibitor treatments, whereas SKO cells are more resistant to such treatments (new **Supplementary Figures 8k** and **l**). Overall, these comprehensive analyses of Lsd1, Foxa2, and AP-1 proteins using GEM mouse models demonstrate consistent findings with those from human PCa models and significantly strengthen the major conclusions in this study.

Reviewer #3 and #4: Comments:

The Ms by Cai and co-Authors entitled “FOXA2-LSD1 epigenetic axis rewires AP-1 for transcriptional reprogramming lineage plasticity in prostate cancer” reports preclinical results that are very interesting to deepen the knowledge on the prostate cancer plasticity that induce cancer cells to progress to CRPC and thereafter into further degeneration, such as the neuroendocrine PCa.

Response: We are so grateful for the reviewer's positive feedback on our study.

Introduction and Discussion are well written and readable, while the Methods, Results, Figures and Legends are very difficult to comprehend, with criticisms in Methods, Results and Figures, regarding in particular experiments done in animals (both in the main Ms and in the Extended data).

Methods are not well described and they do not give enough information, in particular concerning the in vivo experiments. No information on the LSD1 inhibitors (also for the in vitro experiments), no information about cell treatment protocol (10 μ M for 24hours based on...?), no information on animal numbers.

Response: We have modified these sections to improve readability and clarity by including information for LSD1 inhibitors, cell treatment protocol, and animal number information.

Results: Results are too many and too compact into the Figures. Figures have too many inserts (more than 10), often with discrepancies in the citation between Figures, Legend and Results.

Response: In response to this concern, we have expanded our manuscript to include 9 figures instead of the original 7, addressing the issue of overly compacted figures. Additionally, we have thoroughly double-checked the citations for each figure to ensure their accuracy.

- Fig. 2, inserts k and m and Methods (lines 756-759): Experiments with zebrafish embryos

What are the TUE zebrafish lines? Why have they been used, besides AB? How many embryos were injected? What do the numbers in the Figure inserts mean?

Then, the major criticism is related to the Figure reported: Although PC-3 are of metastatic origin, the experimental approach using 100 cells injected in the perivitelline space, an avascular space, taking a picture 1hour after injection is difficult to comprehend. Authors need to explain the discrepancy with other published papers, in which higher number of PC-3 cells and much more longer time-course points are used (i.e. doi: 10.1038/s41388-019-1103-0; 10.1038/s41598-019-44452-4) or, more in general, with the methodological approach generally used for these experiments (reviewed in doi: 10.3390/cells9091978; 10.1016/j.bbdis.2013.01.016). Images are of poor resolution, the shFOXA2 embryo has cells injected in the yolk, not in the perivitelline sac, while pictures of the embryos in the veh and ORY image present a very high background fluorescence, that does not allow to see the cell mass and that could mask the presence of metastasis. Furthermore, the sagittal alignment is not correct, i.e. shFOXA2 embryo displayed both eyes. Finally, a minor point concerns how the zebrafish embryos are usually depicted: indeed, present figures need to be rotated by 180°, as conventionally, zebrafish embryos are shown with the head on the left and the yolk facedown

Response: We appreciate the reviewer's comments. Regarding the zebrafish study, the use of fish embryos, which lack a fully developed immune system, provides a quick method to evaluate cancer cell invasion into

circulation. This approach is gaining recognition as a practical in vivo method for examining cancer cell dissemination and adopted by us in many previous publications (PIMID: 31243372, 37663929, 37549269, 33632899, 38001672). In our experiments, we used the TUE line of zebrafish, which is a commonly used wild-type zebrafish line originally established in Tübingen Germany. We have changed the abbreviation of TUE to the full name "Tübingen". Because both are wild-type lines, we used both AB and Tübingen embryos in these experiments depending on which fish were successfully breeding. We injected approximately 100 GFP-labeled cells (a very common number for this type of study, see reference PMID 32867288) into each zebrafish embryo, with each experimental group comprising 10-20 fish. Following the injection, we conducted repeated imaging of the embryos for up to 24 hours to monitor fluorescence signals within the blood vessels. The presented imaging was performed 1-hour post-injection. We observed no additional invasion activity after this 1-hour mark. The extended duration used in other studies might be attributable to variations in injection techniques (such as the use of Matrigel) or differences in the sites of injection (e.g., yolk sac vs. perivitelline space).

Any embryo displaying signs of circulation signals, even at a single location, was categorized as "invaded". The number in the figure in our manuscript illustrates the percentage of invaded embryos to the total number of embryo injected. The images were taken on a fluorescent dissecting microscope as this gives sufficient resolution to score whether or not cells had migrated out of the injection site. The gain is somewhat high in the picture to allow for visualization of the fish in the image but does not mask any cells that migrated out of the injection site. We were careful to score these accurately on the microscope and provide representative images of what we saw on the microscope. We have revised our methods section and figure legends for the clarification of this process. It is important to note that, in most cases involving prostate cancer cells, we only detected invasion but not the establishment of xenograft tumors.

Regarding the injection sites, we ensured all injections were through the perivitelline space. Any fish accidentally injected into the yolk sac were immediately excluded from the study. Yolk injections tend to have a much more compact sphere of cells at the injection site than what is pictured in the shFOXA2 image. It is also much easier to see the injection location when looking directly at the fish on the scope where we can observe the larva at different angles than it is in the snapshots.

Additionally, because these images were taken of live fish in E3 buffer on the dissecting scope, they are sometimes tilted towards one side and so are not flat on their sides in all images. However, we looked carefully at all fish while on the microscope to find cells outside of the injection site and did not rely solely on snapshots for scoring. we have reoriented all figures in the manuscript to present the head on the left and the yolk facing downward (see revised **Figure 4m-n**). We also modified the methods section to include more information.

- Fig.2j-l, Fig. 4i-j: Experiments with SCID mice and Methods (lines 762-767) Authors needs to clarify the timing of tumor volume measurement, as it seems that after injection of 2×10^6 wt PC-3 cells (very low number compared to published literature), in 6 days (showing an unexpected very high proliferation rate compared to other reports) the tumor volume reaches about 100 mm³ volume (Fig.2j), while in Fig.2l and Fig. 4 I – j, at day 0, the volume is already about 100mm³

Response: For the establishment of PCa cell line-derived xenografts, injecting approximately 1-4 million cells at each site is a standard practice, as used in our many previous publications (PMID375492269, 36877164, 34975152, 37663929, etc.). Particularly, PC-3 cells, known for their high aggressiveness, typically develop xenograft tumors rapidly, often within 6-8 days following injection. In monitoring tumor growth in siNTC versus siFOXA2 cells, we define the day of injection as day 0 (previously Figure 2j, now revised to **Figure 4o**). For experiments involving drug treatment, tumor bearing mice are allocated to different study groups once the average tumor size reaches 100mm³. In these cases, we consider the first day of treatment as day 0 (previously Figure 2l, 4i, 4j, now revised to **Figure 4p, 6j, 6k**). We have elaborated on these procedures in the updated methods section for clarity.

- Many western blot images are of poor quality. No one has been quantified

Response: We have incorporated high-resolution images to improve the quality of the immunoblotting results in our resubmission. We believe the changes shown in all of our blots are quite evident and, therefore, additional quantification might not contribute significant new information. However, should the editors request it, we are prepared to quantify any specific blots.

Minor points:

Why Authors decided to xenograft PC-3 cells in zebrafish embryos to study the metastasis formation and in nude mice to study the tumor volume?

Response: PC-3 cells, which we have characterized as a model of stem-cell-like PCa, exhibit high expression of FOXA2. Our research has clearly demonstrated that in PC-3 cells, FOXA2 binding is regulated by LSD1. Moreover, we have shown that FOXA2 expression leads to the reprogramming of AP-1 signaling. These findings collectively establish PC-3 cells as an ideal model for investigating the effects of FOXA2 silencing, LSD1 inhibition, and the use of AP-1 inhibitors in vivo.

Fig.7: I do understand the significance of CRPC-SCL and CRPC-NE, while what the acronym CRPC-WNT means?

Response: Tang et al. identified the CRPC-WNT subtype of PCa, as detailed in their publication in *Science* (PMID 35617398). This subtype is distinguished by an enrichment of WNT transcription factors TCF/LEF motifs, uniquely present at chromatin accessibility sites specific to this subtype.

Reviewers' Comments:

Reviewer #1:

Remarks to the Author:

Authors conducted additional experiments and have thoroughly addressed all concerns.

Reviewer #2:

Remarks to the Author:

The authors have satisfactorily addressed my concerns.

Reviewer #3:

Remarks to the Author:

I co-reviewed this manuscript with one of the reviewers who provided the listed reports as part of the Nature Communications initiative to facilitate training in peer review and appropriate recognition for co-reviewers.

Reviewer #4:

Remarks to the Author:

I consider acceptable the Authors' responses to the critical issues I raised.

I would just like to point out that in science, an answer like "changes shown in all of our blots are quite evident" is hardly acceptable: defining a result "quite evident" is barely scientific.

Furthermore, not in each wb the bands were yes/no, and bands of different intensity could have been quantified. However, this was a minor point

RE: MS# NCOMMS-23-38567B

“FOXA2 rewires AP-1 for transcriptional reprogramming and lineage plasticity in prostate cancer”

Summary of the Revision: We are delighted that our manuscript has been provisionally accepted. We extend our sincere gratitude to the reviewers and editors for their valuable contributions. In this revision, we have addressed the concerns raised by Reviewer 4 and fulfilled all editorial requests. We eagerly anticipate the publication of our paper in *Nature Communications*.

Reviewers' Comments:

Reviewer #1: Comments:

Authors conducted additional experiments and have thoroughly addressed all concerns.

Reviewer #2: Comments:

The authors have satisfactorily addressed my concerns.

Reviewer #3: Comments:

Authors have conducted additional experiments and all the raised concerns have been extensively addressed.

Reviewer #4: Comments:

I consider acceptable the Authors' responses to the critical issues I raised. I would just like to point out that in science, an answer like "changes shown in all of our blots are quite evident" is hardly acceptable: defining a result "quite evident" is barely scientific. Furthermore, not in each wb the bands were yes/no, and bands of different intensity could have been quantified. However, this was a minor point.

Response: We sincerely appreciate the reviewer's positive feedback on our study and acknowledge that using non-scientific language, such as "changes shown in all of our blots are quite evident," may not be suitable in the authors response letter. As per the request, we have quantified all bands in the immunoblotting gel bots, both in the main figures and supplementary figures. The data are now included in the Resource Data file.